# Bayesian analysis of retinotopic maps

**Noah C Benson[1]\*, Jonathan Winawer[1,2]**

[1]Department of Psychology, New York University, New York, United States; [2]Center for Neural Sciences, New York University, New York, United States

**Abstract** Human visual cortex is organized into multiple retinotopic maps. Characterizing the arrangement of these maps on the cortical surface is essential to many visual neuroscience studies. Typically, maps are obtained by voxel-wise analysis of fMRI data. This method, while useful, maps only a portion of the visual field and is limited by measurement noise and subjective assessment of boundaries. We developed a novel Bayesian mapping approach which combines observation–a subject's retinotopic measurements from small amounts of fMRI time–with a prior–a learned retinotopic atlas. This process automatically draws areal boundaries, corrects discontinuities in the measured maps, and predicts validation data more accurately than an atlas alone or independent datasets alone. This new method can be used to improve the accuracy of retinotopic mapping, to analyze large fMRI datasets automatically, and to quantify differences in map properties as a function of health, development and natural variation between individuals.
DOI: https://doi.org/10.7554/eLife.40224.001

## Introduction

Visual responses in a substantial part of the human brain are organized into retinotopic maps, in which nearby positions on the brain represent adjacent locations in the image. Accurate measurement of these maps using functional magnetic resonance imaging (fMRI) is essential to a wide range of neuroscience and clinical applications (*Wandell and Winawer, 2011*), in which they often provide a basis to compare measurements across individuals, groups, tasks, stimuli, and laboratories. In particular, maps are employed to study homology between species (*Sereno and Tootell, 2005*), cortical plasticity (*Wandell and Smirnakis, 2009*), individual variation in cortical function (*Dougherty et al., 2003*; *Harvey and Dumoulin, 2011*), and development (*Van Essen, 1997*; *Conner et al., 2004*). Many studies of cortical visual function in human, whether in motion (*Huk et al., 2001*), color (*Engel et al., 1997a*), object recognition (*Grill-Spector et al., 1998*), or attention (*Martínez et al., 1999*), include retinotopic mapping as a first step. Finally, basic properties of the maps themselves, such as the cortical magnification function (mm of cortex per degree of visual field), can be used to understand visual performance (*Duncan and Boynton, 2003*).

Despite their broad importance to neuroscience research, no method currently exists to fit a retinotopic map to a subject's cortical surface based on measurement, without human intervention. Rather, most retinotopic analyses of fMRI data use a voxel-wise approach. The general method is (1) to measure responses to mapping stimuli, (2) to derive retinotopic coordinates for each voxel or surface vertex by analyzing traveling waves (*Sereno et al., 1995*; *Engel et al., 1997b*) or by solving a population receptive field (pRF) model (*Dumoulin and Wandell, 2008*) for each voxel, and (3) to identify areal boundaries by visual inspection. Aside from requiring significant time and effort, the maps that result from this process retain many common sources of error including distortion of the BOLD signal due to partial voluming (*Dukart and Bertolino, 2014*), vessel artifacts (*Winawer et al., 2010*), other sources of physiological noise, and model fitting biases (*Binda et al., 2013*). Due to the various sources of noise, the measured maps have discontinuities and often systematically miss portions of the visual field, such as the vertical meridian (*Silver et al., 2005*; *Larsson and Heeger, 2006*; *Swisher et al., 2007*; *Arcaro et al., 2009*; *Mackey et al., 2017*). Further, the measured maps

\*For correspondence: nben@nyu.edu

Competing interests: The authors declare that no competing interests exist.

are limited by the available stimulus field of view in the scanner, often as little as 6–12° of eccentricity, and have difficulty measuring the foveal representation (*Schira et al., 2009*), the portion of the maps most important for many visual tasks including reading and object recognition (*Malach et al., 2002*). These many shortcomings of the traditional retinotopic mapping process derive from the fact that it is organized around optimizing the explanatory power of the retinotopy solutions from individual voxels, rather than that of the entire visual field or cortical area. As a consequence, it yields maps that are neither smooth nor complete—nor grounded in any context of how the visual field is warped onto the cortical surface. Lacking these data, the comparison of maps between subjects is difficult, and precise quantitative examination of individual differences is impossible. We refer to retinotopic maps predicted using voxel-wise methods as being derived from 'Data Alone' (*Figure 1*) because the pRF parameters of the individual voxels come from empirical measurements but are not contextualized in a model of retinotopic maps.

An alternative to voxel-wise modeling of fMRI data is to build a retinotopic atlas—a computational model of the mapping between visual field position and cortical structure. Atlases are typically fit to a group-average description of function on the cortical surface after inter-subject cortical surface co-registration (*Dale et al., 1999*; *Fischl et al., 1999a*). An example group-average description of retinotopy from the Human Connectome Project (*Uğurbil et al., 2013*; *Van Essen et al., 2013*; *Benson et al., 2018*) and the corresponding atlas description are shown in *Figure 2A and B*. Such descriptions are useful despite large inter-subject variation because co-registration of the surface anatomies between subjects improves the inter-subject alignment of cortical function as well. For example, the surface area of V1 can vary by 2- to 3-fold across healthy adults (*Dougherty et al., 2003*; *Stensaas et al., 1974*; *Andrews et al., 1997*), yet atlases of the mean anatomical locations (*Wang et al., 2015*) and the mean functional organization (*Benson et al., 2012*; *Benson et al., 2014*) of V1 can predict the functional organization of left-out subjects with high accuracy. The atlas, after being fit to training data, is applied to an individual anatomical MR image without functional data via anatomical alignment of the image to the atlas followed by interpolation (*Figure 2C*). These atlases solve two of the problems of voxel-wise retinotopic maps: they represent the entire visual

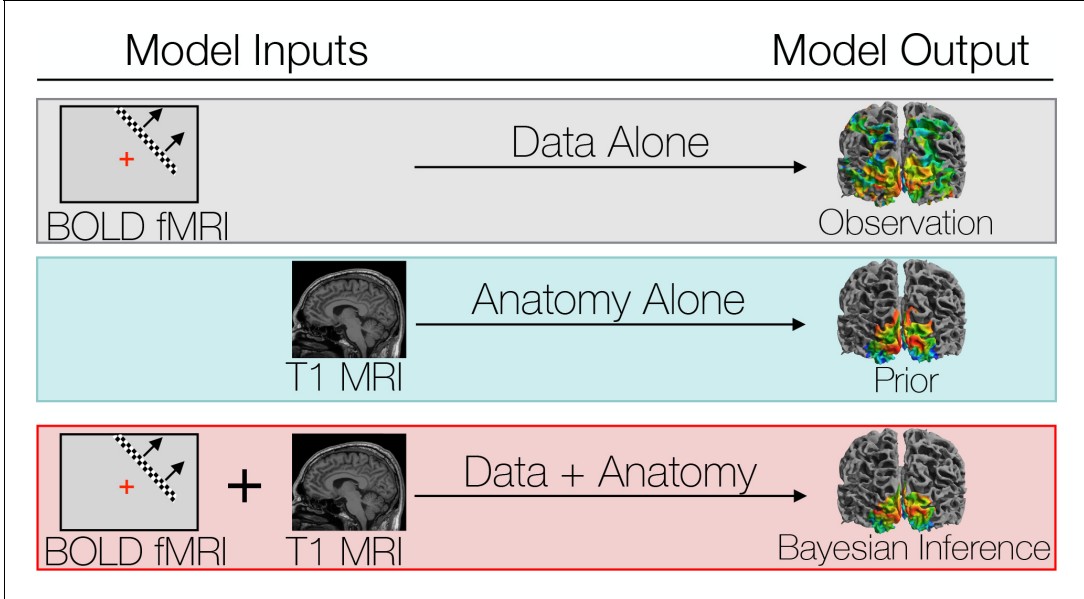

**Figure 1.** We compare three different ways to predict a subject's retinotopic maps. The first method is to perform a retinotopic mapping experiment. The fMRI measurements are converted to retinotopic coordinates by a voxel-wise model and projected to the cortical surface. Although a model is used to identify the coordinates for each vertex or voxel, we call this 'Data Alone' because no spatial template of retinotopy is used. The second method is to apply a retinotopic atlas to an anatomical scan (typically a T1-weighted MRI) based on the brain's pattern of sulcal curvature. This is called 'Anatomy Alone' because no functional MRI is measured for the individual. The third method combines the former two methods via Bayesian inference, using the brain's anatomical structure as a prior constraint on the retinotopic maps while using the functional MRI data as an observation.
DOI: https://doi.org/10.7554/eLife.40224.002

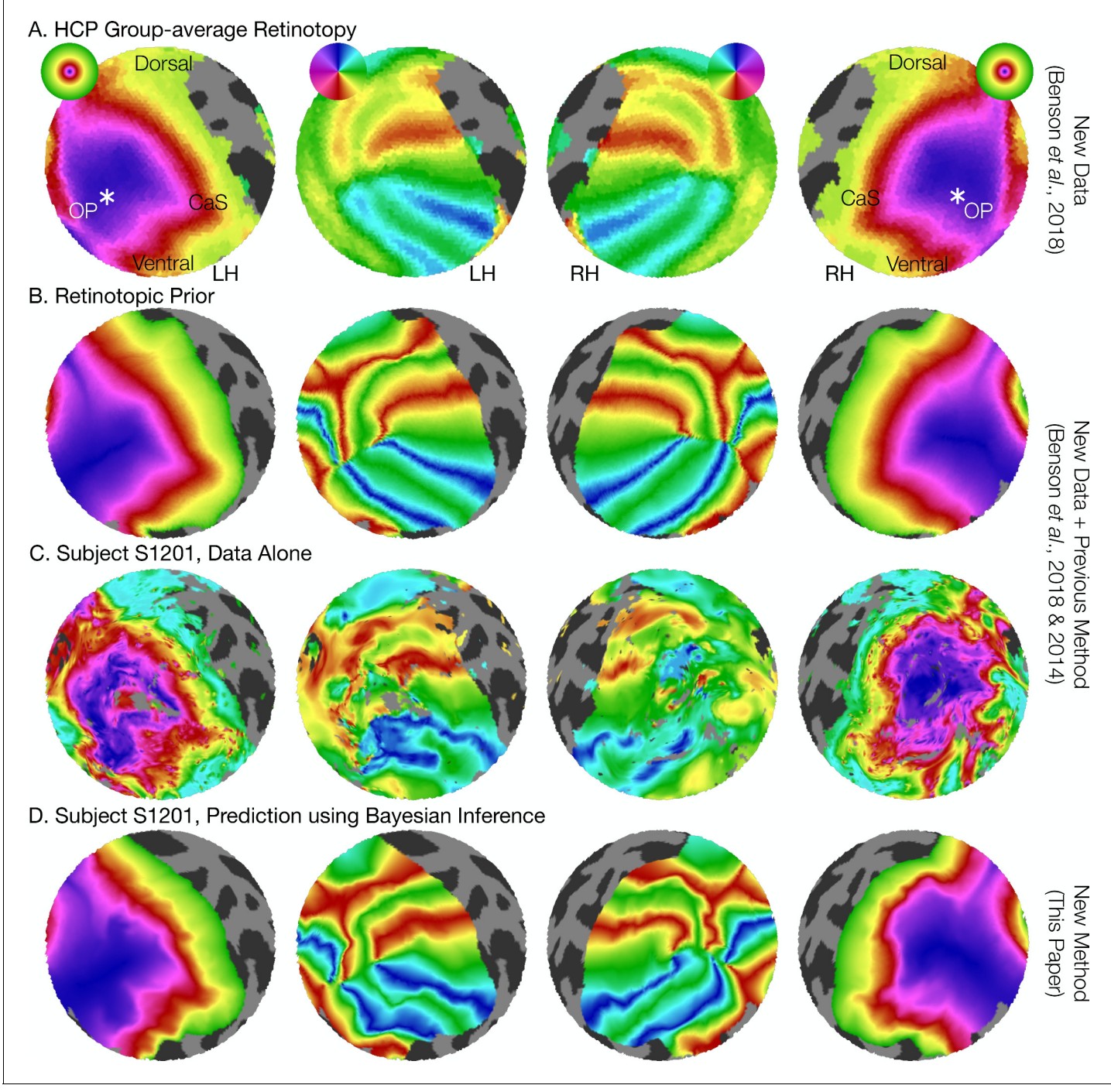

**Figure 2.** The retinotopic prior and its use in predicting retinotopic maps. (A) The retinotopic prior is based on the Human Connectome Project (HCP) group-average retinotopic maps (*Benson et al., 2018*), shown here on an orthographic projection of the V1-V3 region. OP indicates the occipital pole, and CaS indicates the Calcarine sulcus. Projections are identical in each row throughout. (B) The retinotopic prior was designed to resemble the HCP group-average retinotopy, and was further warped to minimize differences between the two according to the methods described by *Benson et al. (2014)*. (C) The measured ('Data Alone') retinotopic maps of subject S1201, all scans combined. Comparison of rows *B* and *C* demonstrates that the use of the retinotopic prior to predict the retinotopic maps of an individual subject results in a reasonable prediction. (D) Combining the retinotopic prior with the observed retinotopic maps from an individual subject yields Bayesian inferred maps.

DOI: https://doi.org/10.7554/eLife.40224.003

field and are smooth, but they are limited by the quality of the anatomical alignment and provide only a description of the mean—they cannot capture the idiosyncrasies of the maps in an individual subject because they assume that once a correspondence is found between the sulcal pattern in two subjects' visual cortices, the function will match. Thus, if one were interested in individual variation in cortical topography *after* anatomical registration, this method is uninformative: it assumes the answer is 0. Accordingly, we refer to retinotopic maps predicted by atlases as being derived from 'Anatomy Alone' (*Figure 1*).

In this paper, we present a solution to the problems of both atlas- and voxel-based retinotopic maps. We hypothesize that a Bayesian model of retinotopic maps, combining data (retinotopic voxel- or vertex-wise measurements) with a prior (a full-field atlas derived from the anatomy), will eliminate many of the issues with retinotopic mapping described above by optimizing the description of cortical retinotopic maps in the context of the full visual field and the corresponding cortex (*Figure 2D*). We propose that such methods can describe cortical retinotopic maps in individual subjects more accurately than an atlas alone or measurements alone. These hypotheses are motivated by two factors. First, previous work employing functional data to supplement global anatomical alignments between subjects has found an increase in the overlap of functional ROIs drawn from independent localizers (*Frost and Goebel, 2013*). Thus, even when subjects are aligned anatomically, appreciable and systematic differences in the structure-function relationship remain. Allowing the measurement from an individual subject to inform the alignment will, in part, capture these individual differences. Secondly, we believe that the basic form of the atlas (the prior) is sufficiently accurate that incorporating it will result in a more accurate estimate of the retinotopic map than the measurements alone.

The method we employ is a Bayesian maximum-likelihood optimization that describes the retinotopic maps in striate and extra striate cortex with previously infeasible precision. Unlike previous work on functional alignment (*Haxby et al., 2011*), we perform alignment between each individual subject's retinotopic parameters and a model of retinotopy described on the (anatomically-aligned) group-average cortical surface. This optimization builds on previous work using iterative approaches to fit and interpolate smooth retinotopic maps in individual subjects (*Dougherty et al., 2003*) by incorporating an explicit prior in the place of human intervention and adopting an explicitly Bayesian formulation.

We publish with this paper a tool capable of implementing the method we describe as well as all source code employed. We use these tools to characterize retinotopic maps from several subjects in terms of the precise warping from visual field to visual cortex. Using these characterizations, we are able to quantify the extent to which variations in retinotopic organization are due to anatomical differences versus differences in the structure-function relationship. We show that, in fact, these two sources of variability—differences in structure and differences in the structure-to-function mapping—are roughly equal and orthogonal across subjects. This means that after warping individual cortical surfaces to bring the anatomies into registration, an additional warping, equal in size, is needed to bring the functional maps into alignment, thereby demonstrating substantial variability in an early, sensory region of the human brain.

## Results and discussion

Retinotopic mapping experiments were performed on eight subjects using fMRI. Twelve individual retinotopy scans were performed on each subject then combined into 21 'training' datasets and 1 'validation' dataset for cross-validation as well as one full dataset of all scans for detailed analysis (*Supplementary file 1*; see also Materials and methods). Predicted maps were then generated using the training datasets and compared to the validation dataset. The training and validation datasets are largely independent in that they are derived from separate scans; however, some dependency remains in that the different scans were obtained from the same scanning session, so that they share anything common to the session (viewing conditions, scanner hardware, etc.). We compared three methods for predicting retinotopic maps (*Figure 1*): (1) using the training data alone as a prediction (the 'observed' maps); (2) using the subject's anatomy alone to apply an anatomically-defined template of retinotopy (the 'prior' maps); and (3) combining data with anatomy (observation with prior) using Bayesian inference (the 'inferred' maps). We then leverage the differences between these methods to characterize the pattern of individual differences in retinotopic maps across subjects.

The prior map (*Figure 2B*), used for methods 2 and 3, was derived from fitting a template to a high-quality dataset derived from 181 subjects in the Human Connectome Project (*Uğurbil et al., 2013*; *Van Essen et al., 2013*; *Benson et al., 2018*) (*Figure 2A*).

## Bayesian inference has the advantages of an anatomical atlas while also respecting individual differences

The inferred retinotopic maps, predicted by Bayesian inference, provide high-quality descriptions of the full retinotopic topology for each subject's V1-V3 regions. These maps can be produced even in the absence of observed retinotopic measurements (i.e., using the prior alone), or by combination with retinotopic data. In this latter case, the inferred maps have all the advantages of the retinotopic prior (topologically smooth maps, predictions beyond the stimulus aperture, etc.), while also accounting for idiosyncrasies in individual maps. Three examples of maps that demonstrate this advantage are shown in *Figure 3*. The first two columns show maps in which, relative to the validation maps, the predictions made from data alone have highly curved iso-eccentricity contours. These contours reflect noise rather than true curvature in the iso-eccentricity contours, as shown by the validation data. For these two columns, the predictions from the prior alone have iso-eccentric contours that are too smooth (as compared to the validation data). The correct lines appear to lie between the training data and the prior. Hence when data is combined with the prior (*Figure 3*, third column) the iso-eccentric contours resemble those of the validation dataset. The third row of *Figure 3* shows an instance in which, even lacking a coherent polar angle reversal to define the ventral V1/V2

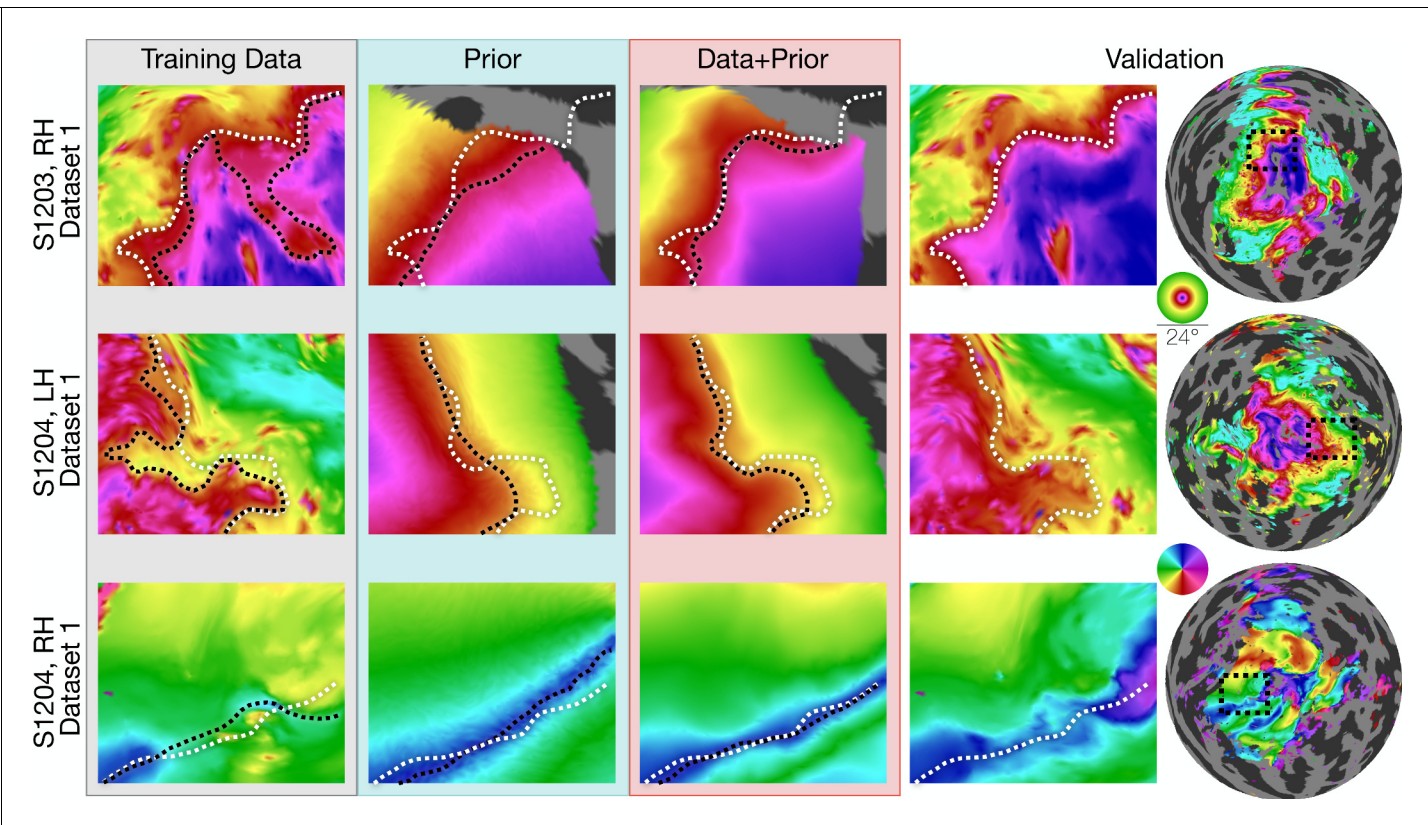

**Figure 3.** Inferred retinotopic maps accurately predict features of validation retinotopy. Twelve close-up plots of the retinotopic maps of three hemispheres are shown with predictions made from Data alone, Prior alone, or Data +Prior. The right two columns show the validation dataset with the right column indicating the context of the close-up patches. The first three columns show different methods of predicting the retinotopic maps, as in *Figure 1*. Approximate iso-eccentricity or iso-angular contour lines for the validation dataset have been draw in white on all close-up plots. Black contour lines show the same approximate contour lines for the three prediction methods. Flattened projections of cortex were created using an orthographic projection (*Supplementary file 2A*).
DOI: https://doi.org/10.7554/eLife.40224.004

boundary near the fovea in the predictions made from data alone, combination of the data with prior more accurately predicts that boundary in the validation dataset than the prior alone.

In constructing the Bayesian-inferred retinotopic map for a single hemisphere, we perform two deformations, detailed in *Figure 4* (see also Materials and methods): (1) we first deform that hemisphere's inflated surface to register it to an average anatomical atlas using FreeSurfer (leftmost arrow in *Figure 4C*); and (2) we then further deform the surface to register it to the retinotopic prior, based on the hemisphere's retinotopic measurements (second arrow in *Figure 4C*). These steps together account for the individual differences in the organization of the subjects' retinotopic maps. In step 1, we account for structural differences across subjects—the deformation that occurs for this registration is unique for each subject. This is where prior work ended (*Benson et al., 2012*; *Benson et al., 2014*). In step 2, we account for the differences in the relationship between structure

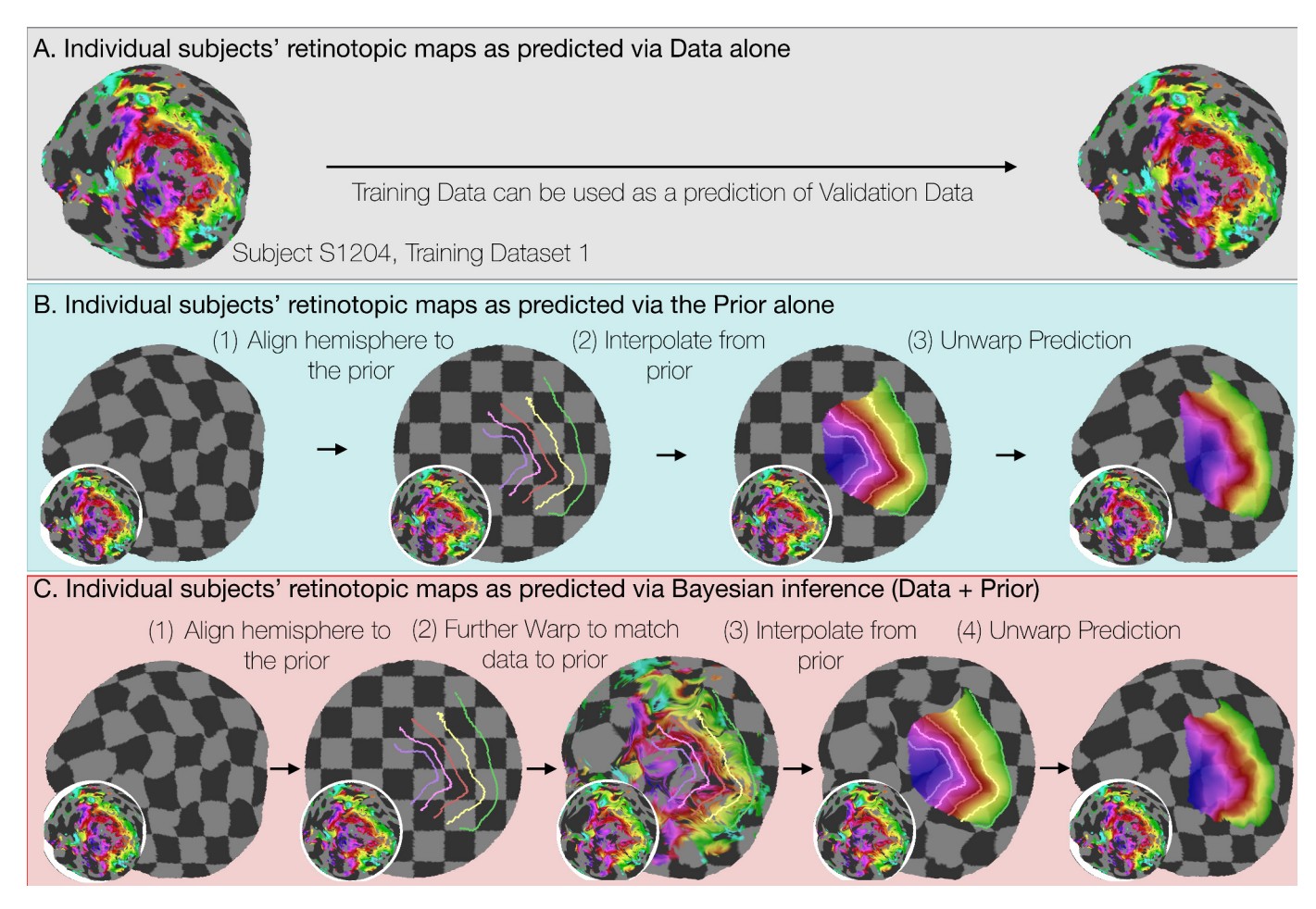

**Figure 4.** Deriving retinotopic predictions. Three methods of predicting retinotopic maps (as in *Figure 1*) for an example subject. (**A**) Predicted retinotopic maps based on training data alone are found by solving the pRF models for each voxel and projecting them to the cortical surface. The training data (left) and prediction (right) are identical. (**B**) To predict a retinotopic map using the prior alone, the subject's cortical surface is aligned to FreeSurfer's *fsaverage* anatomical atlas (represented by rectilinear checkerboards), bringing the subject's anatomy into alignment with the anatomically-based prior, which is represented by iso-eccentricity contour lines in the figure (see also *Supplementary file 2C*). The model of retinotopy is then used to predict the retinotopic parameters of the vertices based on their anatomically-aligned positions. After the predictions have been made, the cortical surface is relaxed. Maps are shown as checkerboards in order to demonstrate the warping (insets show original data and curvature). (**C**) Bayesian inference of the retinotopic maps of the subject are made by combining retinotopic data with the retinotopic prior. This is achieved by first aligning the subject's vertices with the *fsaverage* anatomical atlas (as in **B**) then further warping the vertices to bring them into alignment with the data-driven model of retinotopy (shown as iso-eccentricity contour lines). The warping was performed by minimizing a potential function that penalized both the deviation from from the prior (second column) as well as deviations between the retinotopic observations and the retinotopic model.
DOI: https://doi.org/10.7554/eLife.40224.005

and function *across* subjects. Although it is possible that deformations in step two partly compensate for imperfections in the first two registrations, we propose that, to a first approximation, the deformations applied in the last step indicate meaningful individual differences in the structure-function relationship.

## Individual differences in the V1-V3 structure-function relationship across subjects are substantial

The Bayesian model fitting allows us to parcellate two sources of variation between individuals: differences in surface anatomy (sulcal patterns) and differences in structure-to-function mapping, that is how retinotopic features map onto the surface anatomy. These two sources of variation map approximately to the two deformations in our atlas fitting: differences in surface anatomy are reflected in the deformation used for the surface alignment, and differences in structure-to-function mapping are reflected in the deformation for retinotopic alignment (the Bayesian fit). Note that in our implementation, both alignments are achieved by warping the individual subject's vertices, the former to minimize error in surface curvature, the latter to minimize errors in retinotopic measures. (After the process is complete, the alignments can be reversed, thereby bringing the retinotopic predictions back into the native anatomical space.) Because both the anatomical alignment and retinotopic alignment are computed as changes in the position of surface vertices, it is straightforward to compare the two processes.

There are some subjects for whom there are large differences between the retinotopic atlas defined by the prior and the atlas defined by the Bayesian fit (*Figure 5A and B*). In this example subject, the iso-eccentricity lines in the Bayesian atlas are substantially more compressed along the posterior-anterior axis compared to the anatomical atlas, and the iso-angle lines in V2/V3 are more dorsal compared to the anatomical atlas. Where there are discrepancies, the Bayesian inferred map is more accurate. For example, the polar angles and the eccentricities in the validation data are approximately constant where the Bayesian map predicts iso-angle and iso-eccentricity lines, but not where the prior map predicts them (*Figure 5B*). This indicates that even after anatomical alignment, the retinotopy in this subject differs systematically from the prior. For some other subjects, the two atlases are in closer agreement such that the prior alone is a good fit to the retinotopic data (*Figure 5C*).

To quantify the two types of deformations, we calculated the mean 3 × 3 distance matrix between (1) a vertex's native position, (2) its position in the anatomical alignment (*fsaverage* position), and (3) its 'retinotopic position' after alignment to the retinotopic prior. All vertices in the V1-V3 region within 12° of eccentricity, as predicted by the Bayesian inference on the full dataset (all retinotopy scans combined), were used. We then performed 2D metric embedding to determine the mean deformation steps and the mean angle between them (*Figure 6A*). Overall, the mean deformation distance across vertices is 3.3° ± 0.6° (μ ± σ across subjects) of the cortical sphere for the anatomical alignment and 3.4° ± 0.4° for the retinotopic alignment. The mean angle between these deformations is 83.0° ± 9.6° (note that this last measurement is in terms of degrees of rotational angle rather than degrees of the cortical sphere). The anatomical alignment corresponds to variation in the surface topology and is accounted for in anatomical atlases (*Benson et al., 2012*; *Benson et al., 2014*); the retinotopic alignment corresponds to variation in the structure-function mapping and is accounted for in the Bayesian model. An additional summary measurement of these deformations, the root-mean-square deviation (RMSD) distances across vertices near the occipital pole in a particular retinotopy dataset, provides a metric of the total warping applied during each step of the alignment process for each subject. A summary of this measurement, as well as various other summary statistics is provided in *Table 1*.

The retinotopic deformation distances were not significantly different than the the anatomical deformation distances, which were still substantial; this is true whether one calculates the mean deformation distance over the entire patch of cortex immediately around the occipital pole (shown in the maps in *Figure 3*) or over only the vertices that are predicted to be in V1-V3; *Figure 6* is calculated over the latter of these two ROIs. Note that if the retinotopic deformation distances had been much larger, the prior anatomical atlases would have been less accurate. Had they been close to 0, then the anatomical atlas alone would have been as accurate as the Bayesian model.

We interpret the warping performed to align retinotopic data to the anatomical prior (the retinotopic alignment) as evidence of individual differences in the way in which retinotopy maps to sulcal

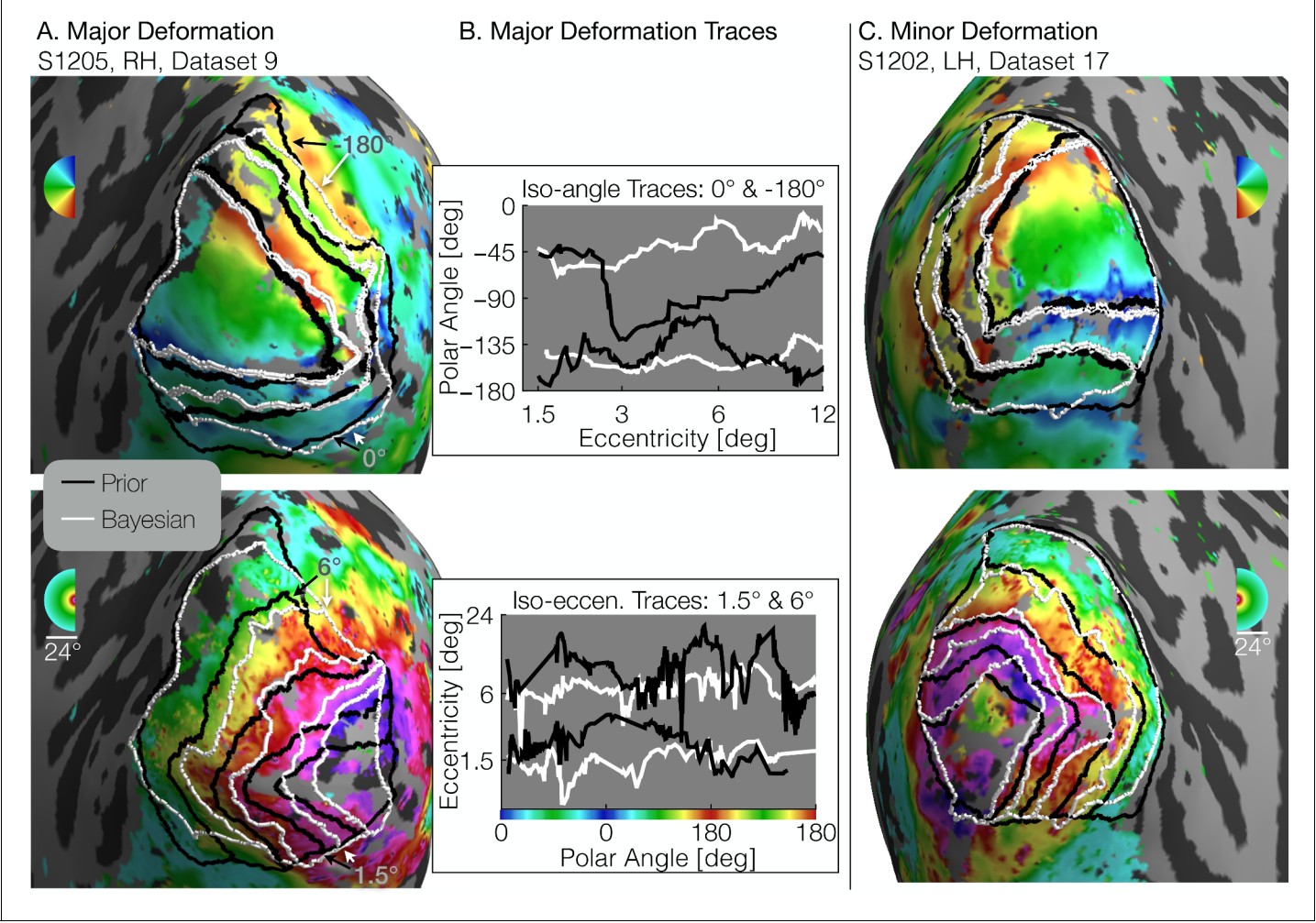

**Figure 5.** Comparison of inferred and prior maps. (**A**) A subject whose maps were poorly predicted by the retinotopic prior and thus required major deformation (S1205, RH, dataset 9). (**B**) To illustrate the differences between the Prior Alone (black lines in *A*) and the combination of Data +Prior (white lines in *A*), traces of the polar angle (top) and eccentricity (bottom) values beneath the lines indicated by arrows are shown. The eccentricities traced by the iso-angle lines and the polar angles traced by the iso-eccentricity lines of the Bayesian-inferred maps more closely match the angles/eccentricities of their associated trace lines than do the polar angles/eccentricities beneath the lines of the Prior alone (**C**) A subject whose retinotopic maps were well-predicted by the prior and thus required relatively minor deformation during the Bayesian inference step (subject S1202, LH, dataset 17). In both **A** and **C**, black lines show the retinotopic prior and white lines show the maps inferred by Bayesian inference.
DOI: https://doi.org/10.7554/eLife.40224.006

topology. An alternate possibility is that the retinotopic alignment corrects for incomplete or incorrect warping performed by FreeSurfer during alignment of each subject's sulcal topology to that of the *fsaverage* hemispheres (the anatomical alignment). We have found FreeSurfer to be a well-vetted and reliable tool for functional alignment; however, no anatomical alignment process is optimal, and more improvements to alignment algorithms, such as those being pursued using the HCP database (*Glasser et al., 2016*), may reduce the length of our retinotopic alignment step. (Note, additionally, that though we use the *fsaverage* here, our tools are compatible with other possible alignments.) It is thus possible that we have overestimated the amount of functional variance remaining across subjects after anatomical alignment. Two observations suggest that the functional variance due to imperfect anatomical alignment is small, however. First, the angle between the alignments is roughly orthogonal, meaning that there is very little movement along the axis of the first (structural) alignment during the second (retinotopic) alignment. Had the first alignment been in the correct direction but too conservative, then we would have expected the retinotopic alignment to be in the same (or similar) direction, rather than orthogonal. Second, if the retinotopic alignment served to correct an

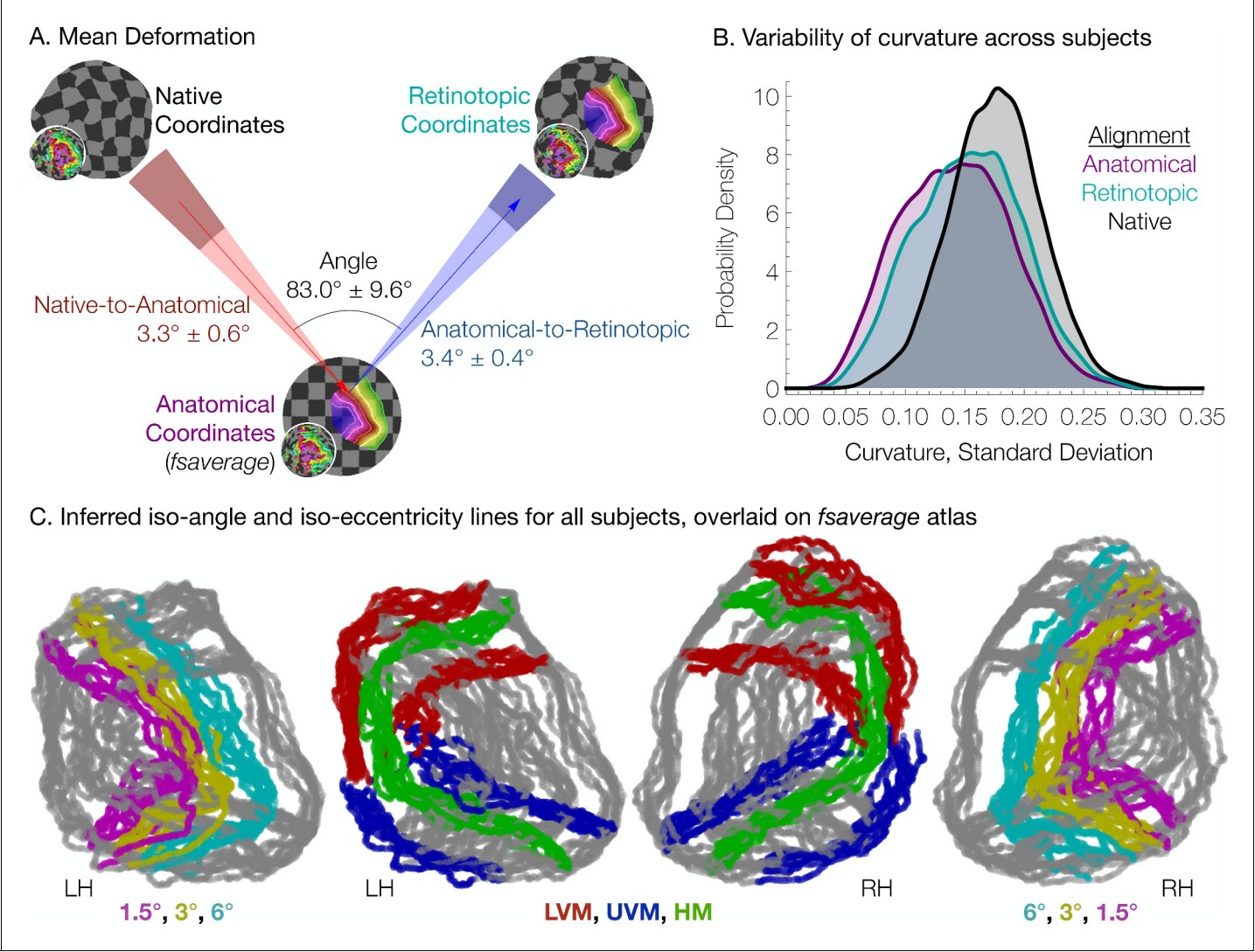

**Figure 6.** Individual differences between subjects in the structure-function relationship are substantial. (A) The mean deformation vectors, used to warp a surface vertex from its Native to its Anatomical (*fsaverage*-aligned) position and from its Anatomical to its Retinotopic position, are shown relative to each other. The wedges plotted beneath the mean arrows indicate ±1 standard deviation of the angle across subjects while the shaded regions at the end of the wedges indicate ±1 standard deviation of the lengths of the vectors. Note that because registration steps are always performed on a subject's inflated spherical hemispheres, these distances were calculated in terms of degrees of the cortical sphere and are not directly equivalent to mm of cortex. (B) The alignment of the V1-V3 region to the retinotopic prior increases the standard deviation of the surface curvature across subjects, suggesting that retinotopic alignment is not simply an improvement on FreeSurfer's curvature-based alignment. Histograms show the probability density of the across-subject standard deviation of curvature values for all vertices in the V1-V3 region with a Bayesian-inferred eccentricity between 0 and 12°. (C) Bayesian-inferred iso-eccentricity lines and V1/V2/V3 boundaries plotted for all subjects simultaneously on the *fsaverage* spherical atlas. Lines are plotted with an opacity of 1/2 to visualize overlap. The left two plots and the right two plots share identical lines but have different colors. Iso-eccentricity lines are colored in magenta (1.5°), yellow (3°), and cyan (6°). Iso-angle lines are plotted in blue (upper vertical meridian), green (horizontal meridian), and red (lower vertical meridian).

DOI: https://doi.org/10.7554/eLife.40224.007

incomplete or incorrect anatomical warping, then anatomical metrics such as curvature would become more uniform across subjects after retinotopic alignment compared to after only anatomical alignment. *Figure 6B* demonstrates that this explanation is unlikely by showing the distribution across surface vertices of the standard deviation of curvature across subjects. When curvature is compared across subjects without retinotopic or anatomical alignment ('Native' alignment in *Figure 6B*) the standard deviation is quite high. When subjects are compared after anatomical alignment, the standard deviations are much lower. After further alignment to the anatomical prior of

**Table 1.** Summary statistics for each subject.

| Subject | Hemisphere | V1 area (mm³)* | V1 volume (mm³)* | Anatomical RMSD† | Retinotopic RMSD† |
|---|---|---|---|---|---|
| S1201 | RH | 1308 | 3733 | 2.88 | 3.15 |
| S1201 | LH | 1315 | 4133 | 1.82 | 2.47 |
| S1202 | RH | 2024 | 3706 | 1.21 | 2.73 |
| S1202 | LH | 2085 | 4199 | 1.28 | 2.65 |
| S1203 | RH | 1574 | 3152 | 2.13 | 3.27 |
| S1203 | LH | 1489 | 2941 | 2.06 | 3.77 |
| S1204 | RH | 1906 | 3325 | 2.29 | 3.00 |
| S1204 | LH | 1645 | 3015 | 2.18 | 3.10 |
| S1205 | RH | 1995 | 3926 | 1.99 | 2.91 |
| S1205 | LH | 1884 | 3372 | 1.76 | 3.31 |
| S1206 | RH | 1647 | 3116 | 2.12 | 2.73 |
| S1206 | LH | 1632 | 2692 | 1.63 | 3.22 |
| S1207 | RH | 1648 | 3402 | 1.84 | 2.41 |
| S1207 | LH | 1421 | 2764 | 1.74 | 3.15 |
| S1208 | RH | 1712 | 3509 | 1.50 | 2.58 |
| S1208 | LH | 1494 | 3083 | 1.89 | 3.08 |

* The V1 boundary was determined from the Bayesian-inferred map constructed by combining the retinotopic prior with the full retinotopy dataset.

† Units of the RMSD values are degrees of the cortical sphere; these are approximately equivalent to mm, but exact measurements in mm are distorted during inflation of the surface. 'Anatomical' RMSD refers to the deviation between the subject's native anatomical sphere and the *fsaverage*-aligned sphere while 'Retinotopic' RMSD refers to the deviation between the *fsaverage*-aligned sphere and the retinotopically aligned sphere. The RMSD values were averaged over all vertices within the inner 12° of eccentricity of the V1-V3 region. Use of a larger patch of cortex (e.g., the flattened map projections in *Figure 4A*) does not qualitatively change the relationship between anatomical and retinotopic RMSD values.

DOI: https://doi.org/10.7554/eLife.40224.008

retinotopy (Retinotopic alignment), the standard deviation of curvature across subjects is between these two extrema. This suggests that the retinotopic alignment is sacrificing some amount of structural uniformity across subjects in order to accommodate the individual differences in subjects' structure-to-function mapping, and is consistent with our interpretation that there are substantial individual differences in the mapping between retinotopy and surface topology.

The large individual differences that remain, even after structural co-registration (*Figure 6C*), point to the importance of using at least some individual subject functional data when inferring the maps, rather than assuming the atlas (prior) is correct. The specific nature of these deformations, and whether, for example, they fall into a few basic patterns, is an important question about the natural variation of individual brains. Our new method, combined with large datasets such as the HCP retinotopy data set (*Benson et al., 2018*) and new alignment tools such as MSMAll (*Robinson et al., 2014*), could be used to address this question.

## The inferred maps make highly accurate predictions with very little data

To quantify the accuracy of our Bayesian-inferred retinotopic maps, and to compare the accuracy against other predictions, we used a cross-validation scheme such that predictions from data alone, the prior alone, or via Bayesian inference were compared against independent validation datasets (*Supplementary file 1*). The validation datasets were derived from 6 of the 12 scans; the predictions from data alone and from Bayesian inference were derived from training datasets, which were comprised of various combinations of 1–6 independent scans (between 3.2 and 19.2 min of data). The predictions from the prior alone did not use training data.

To compare methods of predicting retinotopic maps (*Figure 1*), vertices of interest were identified using the maps inferred from the validation dataset. All vertices from the inner 12° of eccentricity of these maps were compared to their counterparts in the predicted maps. Note that the inferred

maps from the validation dataset were used only to identify the vertices included in the comparison; the retinotopic coordinates from the validation datasets themselves were taken as the 'gold-standard' measurements. In computing prediction accuracy, we weighted the vertices by the fraction of variance explained for each vertex's pRF solution in the validation dataset. For the three types of training datasets (prior alone, data alone, Bayesian inference), we assume that each vertex makes a prediction regardless of the variance explained. To prevent errors at high eccentricity from dominating the error metric, we calculated the scaled error for a vertex to be the angular distance in the visual field between its retinotopic coordinates in the predicted map and the validation dataset, divided by the eccentricity in the validation dataset. *Figure 7* shows the scaled mean squared error (MSE) for the various predicted maps in terms of the amount of time spent collecting retinotopic data for the map.

The maps predicted via Bayesian inference were highly accurate irrespective of the amount of data used to inform the fits (*Figure 7*). Between those inferred maps informed by 3.2 min of scan time (one scan) and 19.2 min (six scans), the scaled MSE of the prediction remains in the range of ~0.4–0.5. These scaled errors are larger near the fovea because the denominator used for scaling the error metric (i.e., the eccentricity) could be very small; when the range is limited to 3 to 12 deg, the MSE is much lower,~0.20–0.26. Expressed separately in units of polar angle and eccentricity, the mean absolute polar angle error from a Bayesian map derived from a single 3.2 min scan is 25° ± 11° and the mean absolute eccentricity error is 0.76° ± 0.34° (μ ± σ across 96 datasets). For the prior alone, these errors are substantially higher: 34° ± 12° for polar angle and 1.3° ± 0.17° for eccentricity. Note that these errors are approximately 3 × higher than those reported for previous versions of the anatomical prior (11° for polar angle and 0.37° for eccentricity) (*Benson et al., 2014*); however, these discrepancies are due to differences in the metric used, the amount of data collected, the thresholding applied, and the use of smoothing. Some of these factors we cannot reproduce exactly (amount of data) or have deliberately abandoned (smoothing), but by using the same metric (median absolute error across all vertices) and similar thresholding (1.25°<predicted eccentricity<8.75°), we obtain errors very close to those previously reported: 5.9° of polar angle and 0.46° of eccentricity. In contrast to the inferred maps for which the accuracy is largely independent of scan time, the accuracy of the predictions from data alone was highly influenced by scan time. The scaled MSE of the maps predicted from the training datasets alone for the same range of scan times ranged from ~2.2 (3.2 minutes of training data) to ~0.3 (19.2 minutes of training data). With more than ~11 min of scan time, the predictions made from the training datasets alone have a slightly lower scaled MSE than

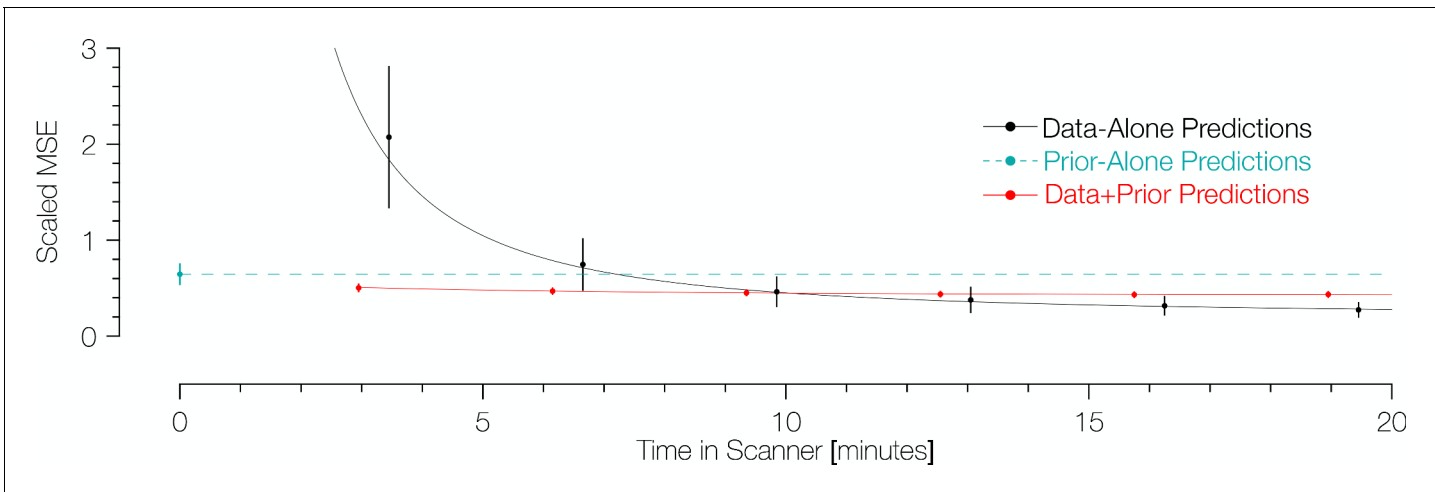

**Figure 7.** Comparison of prediction errors for three methods of predicting retinotopic maps. Errors are shown in terms of the number of minutes spent collecting retinotopic mapping scans (*x*-axis). The *y*-axis gives the mean squared eccentricity-scaled error. Each plotted point represents a different number of minutes in the scanner, with error bars plotting ±1 standard error across subjects. For short scan times, errors are significantly higher for predictions made with the data alone than for those made using Bayesian inference. An offset has been added to the *x*-values of each of the black (−0.25) and red (+0.25) points to facilitate viewing.

DOI: https://doi.org/10.7554/eLife.40224.009

those made from Bayesian inference; although, notably, the improvements in scaled MSE for more than 11 min of scan time are small.

The prediction accuracy from the prior alone (0 min. scan time) was generally intermediate in accuracy between the predictions from the Bayesian model and the data alone. Predictions derived from the anatomically-defined atlas alone are more accurate than predictions from 3.2 min of scanning and only slightly less accurate than predictions from training datasets derived from 6.4 min of scanning; this is in agreement with previous analyses of prediction error versus measurement error in retinotopic mapping experiments (*Benson et al., 2014*). The predictions using the prior alone were universally less accurate than the Bayesian predictions (*Figure 7*, cyan)—for all subjects and all training datasets, the combination of prior with data improved prediction accuracy compared to the anatomical prior alone. These data demonstrate that the application of our new method to a small amount of retinotopic mapping data yields a higher quality retinotopic map than can be derived from other sources alone, with the possible exception of data derived from a long retinotopic mapping session.

The fact that the increased prediction accuracy from the Bayesian maps is almost independent of the amount of scan time used for the observations suggests that much of the individual variability is captured by a low dimensional warping from the template, which can be inferred from a modest amount of data. This hypothesis is further supported by visual inspection of different datasets, such as in *Figure 8A*. Although the amount of noise in the maps clearly varies between the validation dataset (19.2 min. scan time, left column), training dataset 1 (3.2 min., second column), and training dataset 10 (6.4 min., third column), the signal is clear enough that a human expert would likely draw similar boundary lines for each map; our method does as well. Importantly, some warpings are not permitted by the fitting algorithm. For example, the topology of the template is a hard constraint, such that vertices cannot pass through one another. This puts an upper limit on the accuracy of the template: the best predictors of left out data might require a change in topology, which is not permitted. We discuss the significance of these issues in the section subsequent section, 'What is ground truth'.

Another significant advantage of the method is that it eliminates the need for human intervention in the process of delineating retinotopic maps and visual areas. In most studies that require retinotopic mapping data, one or more experimenters hand-label the visual area boundaries. While human raters are better able to understand atypical retinotopic boundaries than our method, they are nonetheless subject to inter-rater disagreement and human error. Furthermore, although expert human raters have a much more nuanced prior about retinotopic map organization than our method, and thus may sometimes draw boundaries better than our method, our method at least makes its prior explicit and quantifiable, and, thus, comparable and replicable across studies.

## What is ground truth?

The motivation for a Bayesian approach to retinotopic mapping can be found most clearly in the measured retinotopic maps themselves. In all of our measured retinotopic maps, there are numerous systematic imperfections (*Figure 8A*), and the literature contains many reports of similar errors (*Winawer et al., 2010*; *Press et al., 2001*; *Gardner, 2010*; *Boubela et al., 2015*). These imperfections can arise from a variety of sources, including partial voluming (*Huettel et al., 2014*), negative BOLD (*Shmuel et al., 2002*), and large, draining veins. Imperfections due to blood vessel artifacts can have effects over large distances (*Winawer et al., 2010*), and most perniciously, they may lead to large and reliable responses that nonetheless differ from the local neuronal activity in the voxel (*Boubela et al., 2015*). Such artifacts can be difficult to track down and are often not eliminated by typical methods of cleaning up maps such as smoothing, thresholding, or simply collecting larger datasets.

Although with large datasets (>19 min of scan time), the prediction accuracy for the validation dataset is highest using the data alone rather than the data and prior, we believe that even in these cases the combination of data + prior is probably closest to ground truth. We defined accuracy operationally as the difference from the validation set, as this provides a single set of independent measures that can be used to assess the accuracy of all three types of models. The validation dataset is defined by at least as many scans as any of the training datasets, and hence is our best measurement. However, the validation dataset is not ground truth, as it is subject to errors from systematic and random measurement noise. The inferred maps, unlike the maps from the validation and training

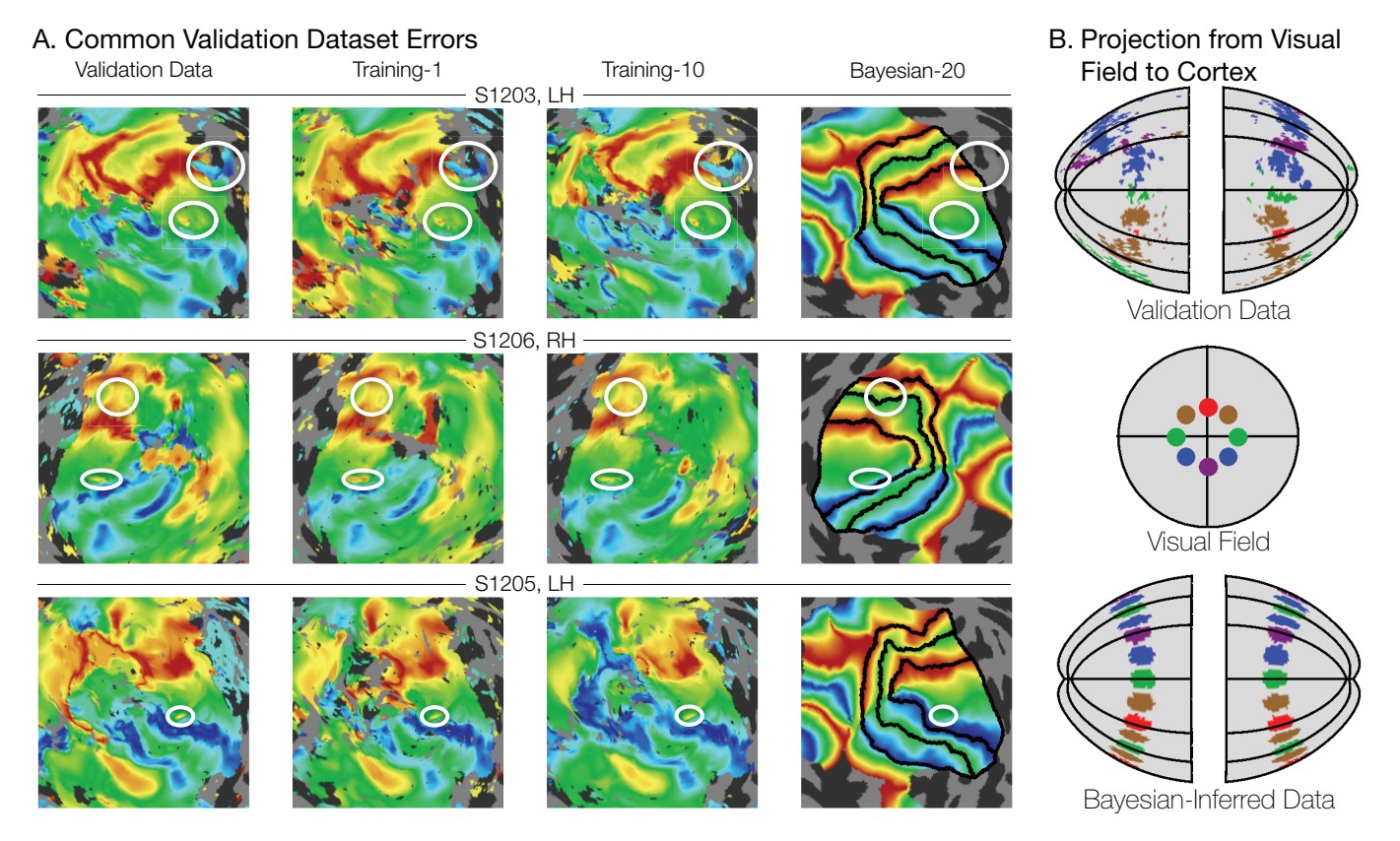

**Figure 8.** Systematic errors in training and validation datasets. (A) Many small inconsistencies in the retinotopic maps are duplicated in both the validation dataset and the training datasets but not in the maps predicted by Bayesian inference. Maps for three example hemispheres are shown with validation datasets as well as training dataset 1, training dataset 10, and the Bayesian-inferred maps from dataset 20. Ellipses highlight blips of noise in the validation maps that are unlikely to represent the true underlying map, but that are correlated with the training maps. Such blips are significantly different in the inferred and validation maps, likely inflating the error of the inferred maps. Black lines show the V1-V3 boundaries in the Bayesian-inferred maps. (B) Discontinuity errors. If the validation data is used to project the disks shown in the visual field in the middle panel to the cortical surface, the resulting map is messy and contains a number of inconsistencies due to measurement error. While the Bayesian inferred map may contain errors of its own, it will always predict a topologically smooth retinotopic map with respect to the topology of the visual field.
DOI: https://doi.org/10.7554/eLife.40224.010

datasets alone, produce a topologically smooth transformation into the visual field: they represent the complete visual field with no holes, redundancies, or discontinuities. Hence, an enclosed region in the visual field will project to an enclosed region in the inferred map on the cortical surface. This is not the case for the maps predicted by the data alone without the use of an atlas (*Figure 8B*). We consider this difference to be an advantage of the inferred maps, since it is generally accepted that the cortical surface of V1 is a topological map of the visual field.

In short, since we do not correct for all of these potential sources of systematic error, we consider our estimates of error from the Bayesian-inferred maps to be conservative, and the estimate from the data-to-data predictions to be liberal.

## The Bayesian model accurately predicts visual field positions not included in the training data

One important advantage of using the method of Bayesian inference outlined in this paper is that it provides predictions beyond the extent of the stimulus aperture in the retinotopy experiment. These peripheral predictions extend to 90° of eccentricity, even though the data used to derive the prior was based on stimuli that only extended to ~8° of eccentricity. Hence, it is important to ask whether the model makes accurate predictions in the periphery. We demonstrate this in two ways. First,

when our registration algorithm is run using only a subset of the eccentricity range (e.g., only data within the first 3° or the first 6° of eccentricity), the predicted maps remain accurate to 12° of eccentricity (*Figure 9A*). Second, we compared wide-field retinotopy data, collected out to 48° of eccentricity from subject S1201 to the Bayesian-inferred map predictions made using our data with a 12°-aperture (*Figure 9B*). We find that in both cases, our method is highly accurate despite lacking training data for peripheral measurements (though note that in the latter case, the extrapolation was only tested on one subject; in principle, subjects with poorer data quality or unstable fixation could result in less accurate extrapolation). Because the extrapolations to untested eccentricities are generally accurate, we conclude that even if prediction accuracy within the measured regions were similar for the Bayesian model and the data-to-data predictions, the Bayesian model is advantageous because it includes predictions for regions of the visual field beyond training data.

## The Bayesian inferred maps accurately reproduce systematic properties of the visual field maps

Another aspect in which our work here extends previous methods is the addition of the pRF size to the retinotopic quantities predicted by the model in the inferred maps—previous work predicted only the pRF centers (*Benson et al., 2012*; *Benson et al., 2014*). Here, we predict the pRF sizes for the vertices based on the eccentricity inferred from the Bayesian map and the assumed linear relationship between eccentricity and pRF size. The inferred pRF size of a vertex is the best linear fit to the measured pRF size versus the vertex's inferred eccentricity (*Figure 10A*). While an approximately linear relationship is reasonable given the literature, the absolute scale is likely dependent on variety of measurement factors such as voxel size, stimulus spatial frequency, and subject fixation (*Alvarez et al., 2015*). Hence, we do not attempt to infer the slope or intercept based on prior measurements.

Another metric inversely related to pRF size is the cortical magnification, usually measured in terms of $mm^2$ of cortex representing one $degree^2$ of the visual field. We summarize these measurements in *Figure 10B and C*. Our measurements of cortical magnification are broadly in agreement with previous work by *Horton and Hoyt, 1991*, shown by the dotted black line panels *B-E* of *Figure 10*. The cortical magnification of the inferred maps is quite similar to that of the observed retinotopic maps. In both cases, V1 has slightly lower cortical magnification than V2 and V3 near the fovea, but higher magnification in the periphery. This difference is slightly exaggerated in the inferred maps relative to the observed maps; although this difference is slight and is in agreement with previous examinations of cortical magnification (*Schira et al., 2009*); however, note that in our maps, the crossover between V1 cortical magnification and V2/V3 cortical magnification occurs at a higher eccentricity (~3°) than previously reported (~0.7–1°). This is emphasized in *Figure 10D and E*, which shows the curves from *Figure 10C and D* in terms of their difference from the prediction of *Horton and Hoyt, 1991* (the black dashed line in panels B-E). Note that in the inferred maps, although the cortical magnification in V1 is lower than V2 below 3° of eccentricity, the difference between them is small between 1.5° and 3°.

## The retinotopic prior and Bayesian-inferred maps include 12 visual areas

Previous research on the retinotopic organization of visual cortex used a model of V1, V2, and V3 retinotopy described by *Schira et al. (2010)* to produce a template of retinotopy that included only those visual areas. This 'banded double-sech' model accurately describes the anisotropic magnification of the visual field on the cortical surface, particularly near the fovea. However, we have observed, particularly in individual data, that retinotopic data from outside the V1-V3 region described by the Schira model has a large impact on the quality of the inferred map. Accordingly, in creating our retinotopic prior, we constructed a new model that includes nine additional visually active regions: hV4, VO1, VO2, V3a, V3b, LO1, LO2, TO1, and TO2. This model employed a new method of constructing 2D models of retinotopy that was specifically designed to accommodate the distortions caused by anatomical alignment, inflation, and flattening of the cortical surface. The new method is much simpler to extend to many more visual field maps, as it does not rely on an analytic description of the flattened (2D) retinotopic maps, which are only available for V1-V3. Rather, it requires as input a cortical map image on which estimates of the visual area boundaries have been

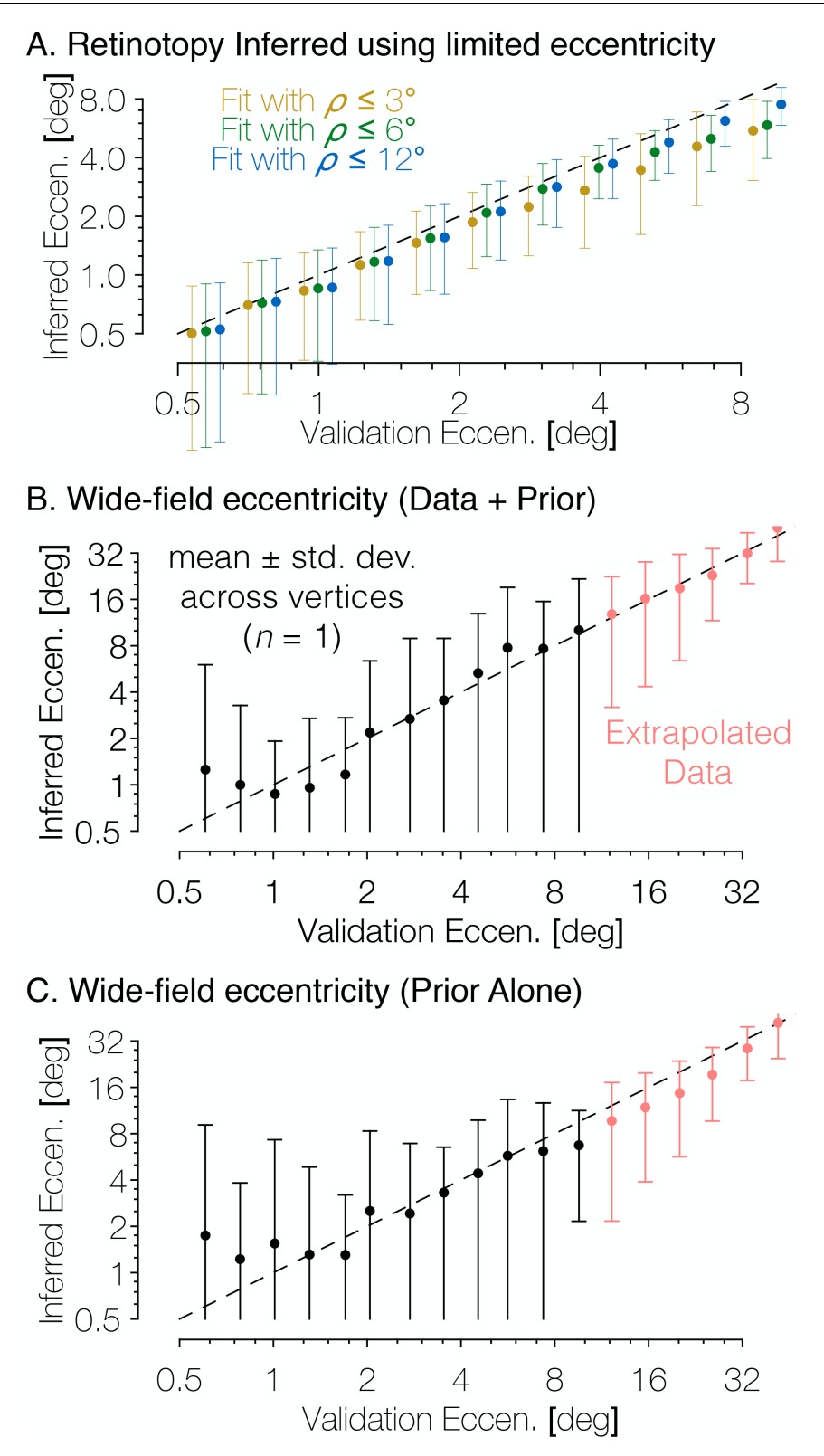

**Figure 9.** The Bayesian-inferred maps accurately predict eccentricity beyond the range of the stimulus. (**A**) In order to examine how accurately the retinotopic maps predicted using Bayesian inference describe the retinotopic arrangement outside of the range of the stimulus used to construct them we constructed maps from all datasets using only the inner 3° or 6° of eccentricity then compared the predictions to the full validation dataset. Eccentricity is well predicted out to 12° regardless of the eccentricity range used to construct the predicted map, indicating that our inferred maps are

*Figure 9 continued on next page*

*Figure 9 continued*

likely accurate beyond the range of the stimulus. In addition, we compared the wide-field retinotopic mapping data from subject S1201 to the inferred retinotopic maps (B) and the anatomical prior (C) using only the 12° stimulus; the inferred eccentricity is shown in terms of the validation eccentricity. The highest errors appear in the fovea (<3°), while predictions made by the inferred maps are most accurate in the periphery, indicating that eccentricity may be well-predicted far beyond the range of the stimulus (out to 48° of eccentricity in this case). Predictions of peripheral data are slightly less accurate when made by the prior than by the inferred maps, which suggests that the extrapolation is improved by the Bayesian inference.

DOI: https://doi.org/10.7554/eLife.40224.011

drawn manually and labeled as either a foveal boundary, a peripheral boundary, an upper vertical meridian, a lower vertical meridian, or a horizontal meridian. A minimization technique is then used to fill in the retinotopic coordinates between the drawn boundaries (see Models of Retinotopy in Materials and methods). The new retinotopic prior, including all new areas can be seen in *Supplementary file 3*. Although we consider the addition of 9 retinotopic areas to be an important development, we consider these areas preliminary and do not analyze them in detail here. One reason for this is that the organizations of many of these areas remain under dispute. Additionally, the responses to our stimuli in these areas is of a considerably lower quality than in V1-V3; thus even were we to analyze the accuracy of the predictions in these ares, our validation dataset would be a particularly poor standard. We do, however, include these areas in the predictions made by the

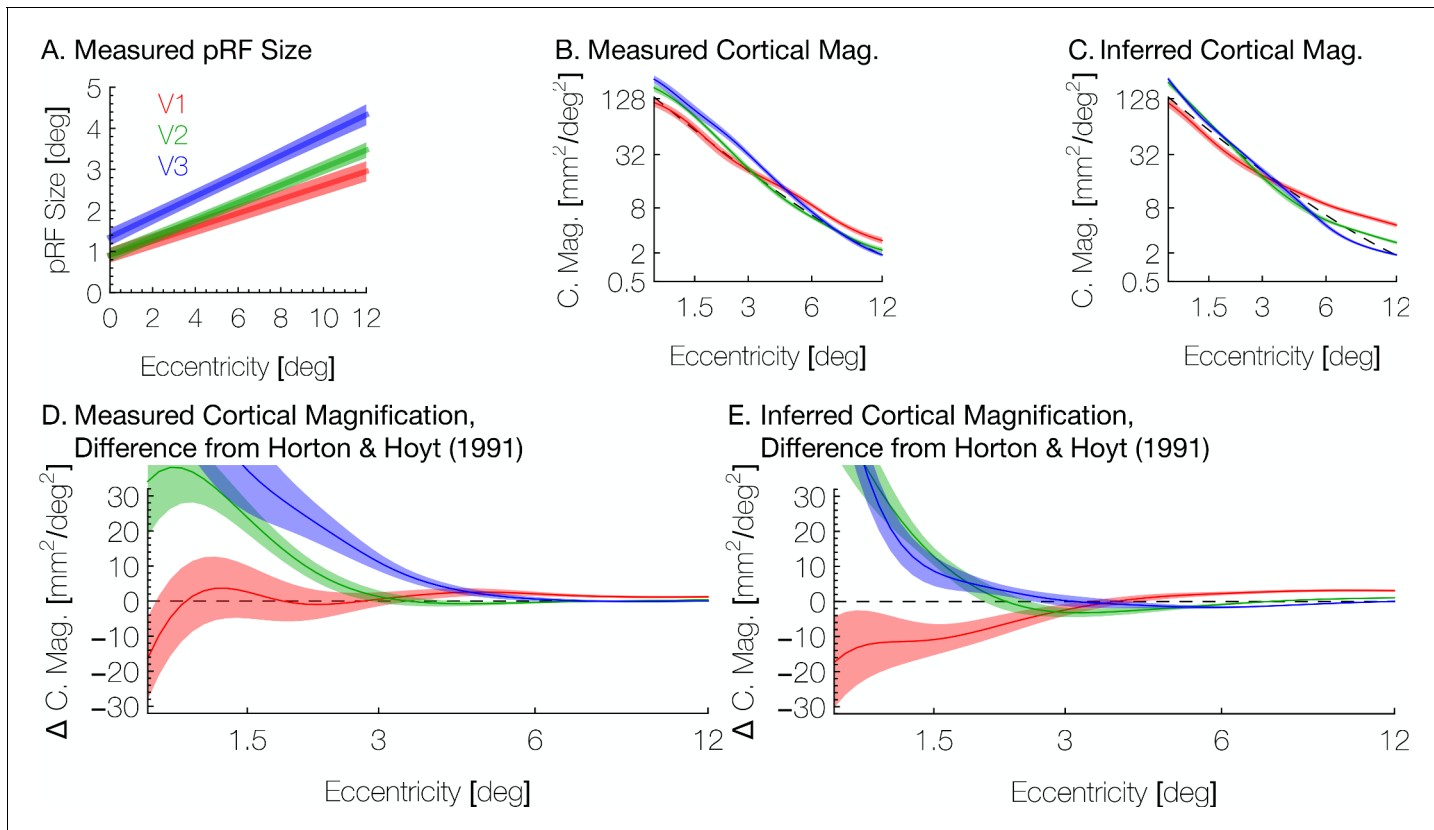

**Figure 10.** Aggregate pRF size and cortical magnification measurements are in agreement with previous literature. (A) PRF sizes by eccentricity are shown for V1, V2, and V3, as calculated from the full datasets; shaded regions show standard errors across subjects. (B) Cortical magnification is shown in terms of eccentricity for V1-V3, as calculated from the full datasets. Again, the shaded regions show standard errors across subjects. The dashed black line shows the equation for cortical magnification provided by *Horton and Hoyt, 1991*. (C) Cortical magnification as calculated using the pRF coordinates inferred by the Bayesian inference. Note that in both *A* and *B*, eccentricity refers to measured eccentricity while in *C*, eccentricity refers to Bayesian-inferred eccentricity. (D) The difference between the cortical magnification predicted by *Horton and Hoyt, 1991* and the cortical magnification of the (D) measured and (E) inferred maps; the data are the same as in *B* and *C*.

DOI: https://doi.org/10.7554/eLife.40224.012

Bayesian inference method so that interested researchers may analyze or use them. These areas are included in the data provided with this paper, and predictions of these areas are included when using the tools provided in the Data Availability Statement.

## Making the Bayesian inference explicit

Our method has the advantage of allowing the retinotopic atlas to act as a prior constraint on new observed data. This is a Bayesian model in the general sense of combining a prior belief with a measurement in order to make an inference. The computation can be formulated in an explicit Bayesian framework. We define a hypothesis $H$ to be a particular warping of the cortical surface, and we define the evidence $E$ to be a particular set of retinotopic measurements. We then convert the cost functions from *Table 2* into probabilities by assuming an exponential relationship. Hence, the prior probability of $H$ is defined in terms of the deviation from the retinotopic prior: $P(H) = \exp\left(-(F_e(x) + F_\theta(x) + F_p(x))\right)$, and the likelihood of the evidence under a given hypothesis, $P(E|H)$, is defined in terms of the fit between the retinotopic model and the retinotopic measurements: $P(E|H) = \exp\left(-(E_e(x) + F_\theta(x) + F_p(x))\right)$. During registration, we seek the hypothesis $H$ that maximizes the posterior probability $P(H|E) = P(E|H)\,P(H)/P(E)$. Because $P(E)$ is a constant, we can ignore it and instead maximize the function given in *Equation 1*, which is equivalent to minimizing $F(x)$. This operation is performed during registration. Thus, to derive our cost function from Bayes' rule, we write:

$$
\begin{aligned}
P(H|E) &= P(E|H)P(H)/P(E) \\
P(H|E) &\propto P(E|H)P(H) \\
P(H|E) &\propto exp\left(-F_\varphi(x)\right) exp\left(-\left(F_e(x) + F_\varphi(x) + F_p(x)\right)\right) \\
P(H|E) &\propto exp\left(-\left(F_\varphi(x) + F_e(x) + F_\varphi(x) + F_p(x)\right)\right) \\
P(H|E) &\propto exp(-F(x))
\end{aligned}
\tag{1}
$$

The explicit Bayesian formulation above clarifies several features of our model. First, the prior

**Table 2.** Components of the registration potential function

| | Term | Description | Form |
|---|---|---|---|
| 1 | $F_e(x; x_0, \mathbf{E})$ | Penalizes changes in the distances between neighboring vertices in the mesh. | $\frac{H_e(x; x_0, \mathbf{E}) + G_e(x; x_0, \mathbf{E})}{|\mathbf{E}|}$ |
| 2 | $F_\vartheta(x; x_0, \Theta)$ | Penalizes the changes in the angles of the triangles in the mesh. | $\frac{H_\vartheta(x; x_0, \Theta) + G_\vartheta(x; x_0, \Theta)}{|\Theta|}$ |
| 3 | $F_p(x; x_0, \mathbf{P})$ | Penalizes any change in the positions of the vertices on the perimeter of the map. | $\frac{1}{2}\sum_{u \in \mathbf{P}}(x)_u - (x_0)_u^2$ |
| 4 | $F_\varphi(x; \Phi)$ | Decreases as a retinotopic vertex $u$ approaches its anchor-point $y$ in the retinotopy model. | $\frac{\sum_{(u,y,\sigma,w) \in \Phi} w \, \exp\left(-\frac{(x)_u - y}{2\sigma^2}\right)}{|\Phi|}$ |
| 5 | $H_e(x; x_0, \mathbf{E})$ | Harmonic component of the edge-length deviation penalty $F_e(x; x_0, \mathbf{E})$. | $\frac{1}{2}\sum_{(u,v) \in \mathbf{E}}(r_x(u,v) - r_{x_0}(u,v))^2$ |
| 6 | $G_e(x; x_0, \mathbf{E})$ | Infinite-well component of the edge-length deviation penalty $F_e(x; x_0, \mathbf{E})$. | $\frac{1}{2}\sum_{(u,v) \in \mathbf{E}}\left(\left(\sqrt{\frac{r_{x_0}(u,v) - q_0}{r_x(u,v) - q_0}} - 1\right)^2 + \left(\sqrt{\frac{q_1 - r_{x_0}(u,v)}{q_1 - r_x(u,v)}} - 1\right)^2\right)$ |
| 7 | $H_\vartheta(x; x_0, \Theta)$ | Harmonic component of the angle deviation penalty $F_\vartheta(x; x_0, \Theta)$. $F_p(x; x_0, \mathbf{P})$ | $\frac{1}{2}\sum_{(a,b,c) \in \Theta}(\alpha_x(a,b,c) - \alpha_{x_0}(a,b,c))^2$ |
| 8 | $G_\vartheta(x; x_0, \Theta)$ | Infinite-well component of the angle deviation penalty $F_\vartheta(x; x_0, \Theta)$. | $\frac{1}{2}\sum_{(a,b,c) \in \Theta}\left(\left(\sqrt{\frac{\alpha_{x_0}(a,b,c)}{\alpha_x(a,b,c)}} - 1\right)^2 + \left(\sqrt{\frac{\pi - \alpha_{x_0}(a,b,c)}{\pi - \alpha_x(a,b,c)}} - 1\right)^2\right)$ |

DOI: https://doi.org/10.7554/eLife.40224.013

probability distributions assumed for vertex lengths are the same for all vertices (rows 1, 5, and six in *Table 2*; P(H) in *Equation 1*). If we had ground truth maps for a large population, we could, in principle, derive edge-specific probability distributions for *Equation 1*, and convert these to edge-edge-specific cost functions (*Table 1*) for the minimization process. We can get a sense of how these distributions might differ across occipital cortex by visualizing the warp fields from our data set (*Supplementary file 1*). These fields show that our registration process causes some vertices to move much more than others, at least in our small subject pool (n = 8). These warp fields are not sufficient to derive edge-specific priors because the number of subjects is small, and because we do not know the end-points of the registration reflect the ground truth maps. The use of a large dataset, such as the 181 HCP subjects (*Benson et al., 2018*) might be helpful in future work to derive edge-specific priors. A further challenge to incorporating realistic priors would be to capture the dependencies across edges in the prior distribution (the joint probability distribution, which would be a function of thousands of variables, one per edge, imposing an enormous computational burden).

A second feature of our method made explicit by the Bayesian formulation is that the prior probability distributions are 0 for solutions that violate the atlas topology. This assumption is implemented implicitly in the cost function, which rises to infinity as the length of an edge approaches 0 or the angle between edges approaches 0. This aspect of the cost function prevents vertices or edges from crossing, thus preserving topology. Because we assume that the cost function is the negative logarithm of the prior probability distribution, the infinite cost indicates an assumed probability of 0. If ground truth data contradicted this assumption (that is, if there were ground truth maps which violated the topology of the model), the prior probability distributions and corresponding cost functions could be changed accordingly.

A third feature of the method is that the likelihood functions depend on the data quality. In *Table 1*, line 4, the weighting of each vertex (w) is proportional to the variance explained by the pRF model. PRF solutions with high variance explained lead to a higher cost when the atlas vertex is far from the corresponding data point. This part of the cost function shows up in the likelihood, $F_\varphi(x)$, in the Bayesian formulation (*Equation 1*). The interpretation is that there is a low likelihood of observing a high-variance-explained pRF solution in a location far from the template solution. A more realistic likelihood calculation (but one that is beyond the scope of our current knowledge and computational resources) would require a noise model that allowed one to compute how likely a pattern of pRF solutions was given a hypothesized map.

## Individual differences in structure-function relationship

An important question in human neuroscience is the degree to which different brains, when brought into anatomical registration, share the same functional mapping. There is no single, agreed-upon method to register the brains of different individuals, but a general finding is that cortical function shows better inter-observer agreement when the brains are aligned based on sulcal topology (surface registration) rather than volume registration (*Wang et al., 2015*; *Van Essen et al., 1998*). Here, using surface registration, we find that substantial individual differences in functional mapping remains, for example as evidenced by the amount of additional warping needed to align individual brains to retinotopic measurements (*Figure 6*). These results are consistent with studies showing differences in structural and functional alignment of primate and human area MT (*Large et al., 2016*). They are also consistent with studies combining structural and functional alignment in the absence of a model or template of the underlying function (*Frost and Goebel, 2013*; *Haxby et al., 2011*). Such studies show that the extra warping driven by functional alignment leads to better predictions of functional responses in cross-validated data.

The anatomical atlas, described previously (*Benson et al., 2014*), was adapted into the first step of our method (*Figure 4C*) and is equivalent to the prediction using the retinotopic prior alone that we present here (*Supplementary file 2B*). Consistent with previous results, we find that the prior alone produces reasonably good predictions for the retinotopic maps of most subjects (*Figure 7*). Additionally, maps predicted using the prior alone contain many of the advantages described here such as topological smoothness, complete coverage of the visual field, and prediction of peripheral retinotopy. However, the idiosyncrasies of individual subjects' retinotopic maps are often not well predicted by the retinotopic prior alone (*Figures 3*, *5A and B*). By combining both the retinotopic prior and a small amount of measured data, we are able to produce higher-quality predictions that

not only share these advantages but also improve the prediction accuracy of the maps beyond that of the measurement error (*Figure 7*).

## Limitations and biases of the inferred maps

We have shown in this paper that the application of Bayesian inference to retinotopic mapping data can yield substantial rewards for basic researchers interested in quantitative retinotopic analysis. However, a number of questions about the scope and limitations of the method remain. For one, it is unclear how our method, developed using data from a small subset of the population (eight subjects), would cope with subjects whose retinotopic organizations are much different than those that are typically assumed in vision science. Such edge-case subjects could include members of clinical populations, such as individuals lacking an optic chiasm (*Hoffmann et al., 2012*; *Bao et al., 2015*; *Olman et al., 2018*), or healthy individuals whose retinotopic boundaries are merely unusual (*Van Essen and Glasser, 2018*). We consider how our Bayesian models perform for edge-cases by fitting the models to two subjects from the Human Connectome Project whose retinotopic maps were recently noted for their peculiarity by *Van Essen and Glasser (2018)*. The retinotopic maps for these subjects as well as the inferred iso-angular and iso-eccentricity lines in V1-V3 are shown in *Supplementary file 5*. Both of these subjects have atypical polar angle organization in their left hemisphere dorsal V2 and V3 maps. Regarding the limitations of the method as applied to subjects such as these, two things are clear: first, the method is unable to reproduce the precise topology of the subjects' unusual dorsal maps, and, second, it is nonetheless capturing most of the maps accurately. In particular, the inferred eccentricity maps are highly accurate despite the mismatched polar angle maps. The polar angle maps cannot be accurately captured by the Bayesian model, because the Bayesian model assumes a prior probability of zero to any solutions that differ topologically from the template, such as these.

In the case of more extreme departures from typical retinotopic organization, such as the overlapping maps observed in achiasmic patients, we cannot be certain that our method would yield coherent inferences. However, we note that this problem is not unique to our method; in fact, generic voxel-wise pRF models also fail for subjects with highly unusual retinotopic organizations such as achiasma. Typically, these model failures are identified, and updates to the model are proposed to account for the relevant conditions. In the case of achiasmic patients, the usual assumption that a pRF can be described as a single Gaussian fails, and new models were developed with two spatially displaced Gaussians (*Hoffmann et al., 2012*). In the case of our method, one might use an alternate formulation of our retinotopic prior in order to better model the maps of the subjects shown in *Supplementary file 5* or subjects from a particular clinical population.

A separate but equally critical question about the method we present is whether it encapsulates any systematic biases about retinotopic organization. Given the field's imperfect knowledge about precise retinotopic organization across individuals, this can be a difficult question to answer; however we note a number of features and assumptions along these lines. With respect to the retinotopic prior, one critical assumptions that was employed during its creation involves the structure of the polar angle reversals at map boundaries (e.g. the V1/V2 boundary or the V3/hV4 boundary). Our prior assumes that the polar angle at these boundaries lies on the vertical meridians (or horizontal meridians in the case of the V2/V3 boundaries). In the group-average retinotopic maps from the 181 HCP subjects (*Supplementary file 3*), however, it is clear that many polar angle reversals occur several degrees away from the vertical (or horizontal) meridian, differing from the prior. In these cases, our Bayesian maps differ systematically from the data. Because the Bayesian computation allows the vertex positions to change but does not allow the retinotopic quantities to change, all retinotopic locations contained in the prior are assigned to some cortical location in the Bayesian map, differing from the observed validation data, which is often missing representations of the vertical meridians. In addition, 'notches' of missing representation of the lower vertical meridian can be seen in several polar angle maps (for example, in *Supplementary file 3A*, black arrows). In both cases—maps that do not quite reach the vertical meridian and maps that have large notches along the boundaries—it is not yet known whether these properties reflect unusual features of the underlying neuronal maps or limits of the fMRI acquisition or analysis.

## Method availability and usage

The method described in this paper has been made publicly available in a number of ways in order to enable easy use by other researchers. The Bayesian inference method itself is implemented as part of a free software library called Neuropythy; we have publicly archived the version of this library used in the preparation of this manuscript at DOI: 10.5281/zenodo.1312983. Additionally, we have created a universally executable Docker image and have publicly archived it at DOI: 10.5281/zenodo.1313859. Detailed instructions for the use of both the library and the universally executable Docker image are available at the Open Science Foundation repository associated with this paper (https://osf.io/knb5g/). In brief, the method may be run with only a few inputs: a FreeSurfer directory for the subject (which can be generated from an anatomical image), and a set of files containing the measured retinotopic parameters for the subject's cortical surface. The outputs produced are a similar set of files describing the inferred retinotopic parameters of the subject's cortical surface. Run-time on a contemporary desktop computer is less than an hour per subject. Detailed instructions on how to use the tools documented in this paper are included in the Open Science Foundation website mentioned above.

# Materials and methods

## Scientific transparency

All source code and notebooks as well as all anonymized data employed in this Methods section and the preparation of this manuscript have been made publicly available at the Open Science Foundation: https://osf.io/knb5g/. Version 0.6.0 and later of the Neuropythy library can automatic download these data and interpret them into Python data structures.

## Subjects

This study was approved by the New York University Institutional Review Board, and all subjects provided written consent. A total of eight subjects (4 female, mean age 31, range 26–46) participated in the experiment. All scan protocols are described below.

## Magnetic resonance imaging

All MRI data were collected at the New York University Center for Brain Imaging using a 3T Siemens Prisma scanner. Data were acquired with a 64-channel phased array receive coil. High resolution whole-brain anatomical T1-weighted images (1 mm$^3$ isotropic voxels) were acquired from each subject for registration and segmentation using a 3D rapid gradient echo sequence (MPRAGE). BOLD fMRI data were collected using a T2*-sensitive echo planar imaging pulse sequence (1 s TR; 30 ms echo time; 75° flip angle; $2.0 \times 2.0 \times 2.0$ mm$^3$ isotropic voxels, multiband acceleration 6). Two additional scans were collected with reversed phase-encoded blips, resulting in spatial distortions in opposite directions. These scans were used to estimate and correct for spatial distortions in the EPI runs using a method similar to (*Andersson et al., 2003*) as implemented in FSL (*Smith et al., 2004*).

Anatomical images were processed using the FreeSurfer image analysis suite, which is freely available online (http://surfer.nmr.mgh.harvard.edu/) (*Dale et al., 1999*; *Fischl et al., 1999a*; *Fischl et al., 1999b*; *Fischl and Dale, 2000*). Subject brains were inflated and aligned to FreeSurfer's anatomical *fsaverage* atlas.

## Stimulus protocols

Each subject participated in 12 retinotopic mapping scans using the same stimulus employed in the Human Connectome Project (*Benson et al., 2018*). Briefly, bar apertures on a uniform gray background swept gradually across the visual field at four evenly-spaced orientations while the subject maintained fixation. Bar apertures contained a grayscale pink noise background with randomly placed objects, faces, words, and scenes. All stimuli were presented within a circular aperture extending to 12.4° of eccentricity. The bars were a constant width (1.5°) at all eccentricities. Subjects performed a task in which they were required to attend to the fixation dot and indicate when its color changed.

The 12 scans were split into several subsets and analyzed as independent datasets. Six of the scans (two of each bar width) were allocated to the subject's *validation dataset*, while the remaining

three scans were used for 21 training datasets: 6 datasets with one scan each, 5 datasets with two scans each, 4 datasets with three scans each, 3 datasets with four scans each, 2 datasets with five scans each, and one dataset with all six training scans. Additionally, all 12 scans were included in a *full dataset* which was used for all analyses not related to the accuracy of the Bayesian inference method.

Additionally, one previously published retinotopy dataset for with a wide field of view (48° of eccentricity) was re-analyzed (*Wandell and Winawer, 2011*) (their *Figure 3*). The subject for this dataset was also included in the newly acquired data, S1201. The wide-field-of-view dataset was used as a further validation set for models derived from the newly acquired data for S1201, as it enabled us to test the accuracy of model predictions in the far periphery from models derived from data with limited eccentricity.

## FMRI processing

Spatial distortions due to inhomogeneities in the magnetic field were corrected using in-house software from NYU's Center for Brain Imaging (http://cbi.nyu.edu/software). The data were then motion-corrected by co-registering all volumes of all scans to the first volume of the first scan in the session. The fMRI slices were co-registered to the whole brain T1-weighted anatomy, and the time series resampled via trilinear interpolation to the 1 mm$^3$ voxels within the cortical ribbon (gray matter). Finally, the time series were averaged for each voxel across all scans with the same stimulus within a given dataset.

## PRF solutions

Retinotopic maps were produced by solving linear, circularly symmetric population receptive field (pRF) models for each voxel using Vistasoft, as described previously (*Dumoulin and Wandell, 2008*). pRF models were solved using a two-stage coarse-to-fine approach on the time series in the 1 mm$^3$ gray matter voxels. The first stage of the model fit was a grid fit, solved on time series that were temporally decimated (2×), spatially blurred on the cortical surface using a discrete heat kernel (approximately equal to a Gaussian kernel of 5 mm width at half height), and subsampled by a factor of 2. The decimation and blurring helps to find an approximate solution that is robust to local minima. The parameters obtained from the grid fit were interpolated to all gray matter voxels and used as seeds for the subsequent nonlinear optimization. Finally, the pRF parameters were projected from the volume to the cortical surface vertices for white, mid-gray, and pial surfaces using nearest-neighbor interpolation; values were then averaged across the three layers using a weighted-mean in which the fraction of BOLD signal variance explained by the pRF model was used as a weight. All vertices with a pRF variance explained fraction less than 0.1 were ignored.

## Models of retinotopy

To generate our initial models of the retinotopic maps, we begin by hand-drawing boundaries for 12 retinotopic maps. These boundaries need only be drawn once for a single group-average retinotopic map. The boundaries are projected onto the cortical surface, and the retinotopic coordinates for each vertex on the surface are deduced via a minimization procedure. This minimization is motivated by two principles: (1) the retinotopic fields (polar angle and eccentricity) should be as orthogonal to each other as possible and (2) the retinotopic fields should be as smooth as possible. To this end, the minimization routine simultaneously maximizes both the smoothness of the retinotopic fields between vertices connected by edges in the mesh as well as the overall orthogonality between the polar angle field and the eccentricity field. The hand-drawn boundary values are held constant during the minimization. *Equation 2* gives the function $f$ that is minimized, where $\theta$ and $\varrho$ represent vectors of the polar angle and eccentricity values, respectively; $\mathrm{E}$ represents the set of edges between vertices in the mesh; and $\mathrm{x}$ represents the matrix of vertex coordinates (i.e., $\mathrm{x}_u$ represents the coordinate vector of the vertex *u*). During minimization the values of the polar angle and eccentricity vectors are scaled such that both fields ranged from −1 to 1 (e.g., polar angle boundary values of 0° and 180° were assigned values of −1 and 1, respectively) so that the fields could be evaluated easily for orthogonality; after the minimization, the polar angle was linearly rescaled back to the range 0°−180° while eccentricity was rescaled so as to have an exponential distribution that best fit the group-average. We employed this model generation routine using boundaries drawn over the

group-average data (see Anatomically-defined Atlas of Retinotopy: Group Average, below) as well as the *Wang et al. (2015)* atlas and the *Hinds et al. (2008)* V1 boundary as rough guidelines.

$$f(\theta, \rho) = (\theta \cdot \rho)^2 + \sum_{(u,v) \in \mathbf{E}} \frac{(\theta_u - \theta_v)^2 + (\rho_u - \rho_v)^2}{2||(x)_u - (x)_v||} \qquad (2)$$

A full description of the model, including how it can be projected onto an *fsaverage* spherical surface or an individual subject's *fsaverage*-aligned spherical surface, is provided in the open source Neuropythy library (https://github.com/noahbenson/neuropythy).

## Anatomically-defined atlas of retinotopy

Construction of the anatomically-defined atlas of retinotopy is summarized in *Supplementary file 2*. Previous work employed a mass-spring-damper system combined with a nonlinear gradient-descent minimization in order to register group-average retinotopic data, averaged on FreeSurfer's *fsavera-ge_sym* hemisphere (*Greve et al., 2013*), with a model of V1, V2, and V3 retinotopy (*Schira et al., 2010*). In this paper, we modify this technique slightly to bring it more in line with previous estab-lished methods such as those used by FreeSurfer for surface-based anatomical registration (*Dale et al., 1999*; *Fischl et al., 1999a*). In brief, retinotopy is measured in a group of subjects via fMRI; the subjects' cortical meshes are aligned to the *fsaverage* surface via FreeSurfer's surface reg-istration; the retinotopic coordinates are then averaged across subjects at each vertex on a single atlas of the cortical surface; a 2D atlas of retinotopy is then placed on this cortical surface; and finally, the cortical surface is warped to match the retinotopic atlas as best as possible given con-straints on the warping. Each of these steps is described in more detail below.

*Group-average Data.* Group-average retinotopic maps (*Supplementary file 2B*) were obtained from 181 subjects whose data were published and made freely available as part of the Human Con-nectome Project (*Benson et al., 2018*). The resulting group-average retinotopic maps are shown in *Supplementary file 2B*.

*Cortical Map Projection.* The cortical surfaces of the *fsaverage* left and right hemispheres, on which the group-average data were constructed, were inflated both to a smooth hemisphere (Free-Surfer's 'inflated' surface) as well as to a sphere (FreeSurfer's 'sphere' surface); the vertices on the spherical surfaces were then flattened to 2D maps using an orthographic map projection. Precise parameters of this projection and the source code used to generate it are included in the Data Avail-ability Statement. We refer to the 2D vertex coordinates in this resulting map as the 'initial vertex coordinates' because they precede the warping of the vertex coordinates that occurs during registration.

*Registration.* The initial vertex coordinates of the map projections described above were warped in order to bring the polar angle and eccentricity measurements of the vertices into alignment with the 2D model's predictions of retinotopy while maintaining topological constraints: that is prevent-ing triangles in the triangle mesh representing the 2D cortical map from inverting and penalizing excessive stretching or compression of the map. This process was achieved by minimizing a potential function defined in terms of the edges of the triangle mesh, the angles of the triangle mesh, and the positions of the vertices with polar angle and eccentricity measurements above the weight threshold (see *Group-Average Data*, above). *Equation 3* gives this potential function, $F(x)$, which is further broken down into four components detailed in *Table 2*. Fundamentally, the potential function $F$ is a sum of two kinds of penalties: penalties for deviations from the reference mesh and penalties for mismatches between the vertices with retinotopic coordinates and their positions in the retinotopic model. In the case of the former, the reference mesh is gi $x_0, E, \Theta, P,$ and $\Phi$ and the potential of the deviations are defined by $f_e$, $f_\theta$, and $f_p$. The latter is described by $f_\varphi$. In these functions, $\mathbf{x}$ represents the $n \times 2$ matrix of the 2D-coordinates of each vertex while $\times_0$ represents the same coordinates in the reference mesh; $\mathbf{E}$ represents the set of undirected edges (represented as $(u, v)$ pairs such that $(u, v)$ and $(v, u)$ are not both in $\mathbf{E}$) in the reference mesh; $\Theta$ represents the set of angle triples ($a, b, c$) such that the angle is between edge $(a, b)$ and edge $(a, c)$; $\mathbf{P}$ is the set of vertices that lie on the perimeter of the 2D map projection; $r_{\mathbf{x}}(u,v)$ is the Euclidean distance between vectors $(\mathbf{x})_u$ and $(\mathbf{x})_v$; and $\alpha_x(a, b, c)$ is the counter-clockwise angle between vectors $((x)b - (x)a)$ and $((x)c - (x)a)$; $\Phi$ repre-sents the set of anchors defined by the retinotopic model in which each anchor is represented by a tuple $(u, \mathbf{y}, \sigma, w)$ where $w$ is the weight of the anchor, $u$ is the vertex drawn to the anchor, $\sigma$ is the

standard deviation of the anchor's Gaussian potential well, and $y\mathbf{y}$ is the 2D point to which the anchor is attached; the constants $q_0$ and $q1$ are the minimum and maximum allowable edge lengths, respectively.

$$F(x; x_0, \mathbf{E}, \Theta, \Phi, \mathbf{P}) = F_e(x; x_0, \mathbf{E}) + F_\vartheta(x; x_0, \Theta) + F_p(x; x_0, P) + F_\varphi(x; x_0, \Phi) \qquad (3)$$

The term of the potential function devoted to the retinotopic model is given in $F_\varphi$ (*Equation 3*; *Table 2*). This potential term is a set of inverted-Gaussian potential wells called anchors. Each anchor represents the attraction of a single vertex $u$, with measured polar angle $\theta$, eccentricity $\varrho$, and weight $w$, to a 2D point $\mathbf{y}$, at which the retinotopic model predicts a polar angle value of $\theta$ and an eccentricity value of $\varrho$. Note that each visual area represents every point $(\theta, \varrho)$ in the visual field, there are multiple anchors per vertex with retinotopic data. In fact, the retinotopic model used in this paper defines nine maps in addition to the V1-V3 maps (see Model of Retinotopy, above), bringing the total number of anchors per retinotopic vertex to 12. The additional areas are intended partly to prevent vertices immediately outside of V1-V3 from being drawn incorrectly into the V1-V3 section of the model and are not analyzed in detail in this paper. Each anchor additionally defines a parameter $\sigma$; this value is the width (standard deviation) of the anchor's Gaussian potential well; $\sigma$ is defined as the minimum distance from the given anchor to any other anchor to which $u$ is also attracted; this value was given a maximum value of $20\varepsilon$ where $\varepsilon$ is the mean edge-length in the projected map.

The potential function was minimized using a gradient descent algorithm sensitive to the singularities in the terms $G_e$, and $G_\theta$ (*Table 2*); whenever the singularity is accidentally crossed, the minimizer backtracks and chooses a smaller step-size. This approach prevents the inversion (from counter-clockwise ordering to clockwise ordering) of any triangle in the mesh, as such an inversion would require the minimization trajectory to pass through a singularity at the point where $\alpha = 0$ or $\alpha = \pi$. The source code used to minimize the potential function as well as specifications of the gradients of each term is provided in the open-source library included with the Neuropythy and Neurotica libraries (https://github.com/noahbenson/nben).

Minimization was run for at least 2500 steps in which the step-size was constrained such that the displacement of each vertex in each step was at most $1/50^{\text{th}}$ of the average edge-length in the map projection. A small amount of exponentially distributed random noise was added to the gradient at each step with the constraint that the gradient direction at each vertex be conserved; this noise did not affect the minimum obtained by the search but did speedup convergence significantly (see associated libraries for further details). Convergence was generally observed within 1000–2000 steps. The set of vertex coordinates that resulted from this minimization brings the retinotopic measurements associated with the vertices in V1, V2, and V3 referred to as the *registered vertex coordinates*.

*Prediction.* The registered vertex coordinates, once obtained, give the alignment of the subject's cortical surface to the model of retinotopy; accordingly, a prediction of any vertex's associated pRF and visual area label can be derived by comparing the the vertex's registered coordinates with the model. Every vertex whose registered coordinates fall within the model's V1 boundary, for example, is labeled as part of V1. Because only the vertex coordinates, and not the vertex identities, are changed during the registration process, there is no need to invert the registration: visual area label, polar angle, and eccentricity values assigned to each vertex apply as readily to the vertices whether they are visualized in the registered vertex coordinates or in the coordinates that define the subject's white-matter surface, for example. The retinotopic map predictions for the group-average data is shown in *Supplementary file 2D* (left column).

Because the group-average retinotopic data were used in the registration, the predicted map that results provides a reasonable estimate of any subject's expected retinotopic map, as shown previously (*Benson et al., 2012*; *Benson et al., 2014*); although the predicted map does not account for further individual differences in the structure to function relationship, as we show in this paper. Additionally, because the predicted map from the group-averaged data is defined on the *fsaverage* subject's cortical surfaces, a retinotopic map prediction for any new subject, for whom retinotopic mapping measurements may not be available, can be easily obtained: one can use FreeSurfer to align the new subject's cortical surface with the *fsaverage* subject's surface (anatomical structure alignment) then to project the retinotopic maps from the *fsaverage* subject to the new subject based on the anatomical similarity between them. Because of this, we refer to this group-average

retinotopic prediction as the *anatomically-defined atlas of retinotopy*. This atlas is used as *the prior* for the Bayesian model fit, described below. The atlas is similar but not identical to one presented previously (*Benson et al., 2014*).

## Bayesian retinotopic maps

The anatomically-defined atlas of retinotopy, while providing a good prediction for most subjects' individual retinotopic maps, nonetheless does not account for individual differences in the mapping between anatomical location and retinotopic coordinates. Accordingly, predicted retinotopic maps for individual subjects were refined starting from the anatomically-defined atlas of retinotopy using a similar method as was used to generate the atlas originally; this process is detailed in *Figure 4C*. For each subject, their cortical surface was aligned based on anatomical structure to the *fsaverage* subject's cortical surface using FreeSurfer, then their retinotopic data were projected to a map using the identical map projection described above in the section on the anatomically-defined atlas of retinotopy. Note that, in this case, the anatomical alignment to the *fsaverage* subject serves to make the map projections as similar as possible between individual subjects and the anatomically-defined atlas of retinotopy. If we were not interested in incorporating information obtained from the anatomically-defined atlas of retinotopy (which represents a prior belief of retinotopic organization based on group-average data), this step would not be necessary.

The individual subject's projected map is then arranged according to the registered vertex coordinates from the anatomically-defined atlas of retinotopy; this step reflects the prior belief that the group-average registration to the retinotopy model is generally accurate for an individual subject when that subject's anatomical structure has been aligned to the *fsaverage* subject's. Critically, none of the steps taken so far in processing the individual subject's data relies on any measurements of retinotopy that might be associated with that subject. Rather, these steps have relied only on anatomical structure. If, for a subject, no retinotopic measurements are made, then there is no data with which to modify this prior belief; accordingly, the prediction of retinotopy for that subject would be identical to the prediction of retinotopy contained in the anatomically-defined atlas. In other words, without observation, the prior remains the prediction.

The next step registers the individually measured retinotopy data to the anatomically-defined atlas. Before registration, the individual subject's data is resampled onto a uniform triangular mesh, and each vertex whose retinotopic measurements are above threshold are given a weight, $w$, based on the variance explained, $\omega$, of its pRF model solution. The mesh is resampled to the same uniform triangle mesh used as the initial vertex coordinates in the registration of the anatomically-defined atlas of retinotopy in order to speedup registration. Triangles that are tightly pinched (i.e., triangles with internal angles near 0 or $\pi$) can drastically slow the registration progress by forcing the minimizer to frequently backtrack steps; resampling makes such behavior much less likely during the initial minimization. Aside from the weight, other parameters tracked by the potential field, including anchors parameters used by the function $F_\varphi(\boldsymbol{x})$, are obtained identically as with the anatomically-defined atlas of retinotopy. These anchors inherit the weight of the vertex to which they apply, but are reduced when the field sign of the triangles adjacent to the vertex does not match the field sign of the visual area to which it is tied by the anchor or when the pRF size predicted by the model does not match that of the vertex's measured pRF. Details regarding the weights on anchors are provided in the neuropythy library.

For each training dataset of each subject, minimization was run for 2500 steps using the same protocol that was used with the anatomically-defined atlas of retinotopy. Retinotopic map prediction, based on the positions of the registered vertex coordinates in the retinotopy model, were also computed identically to those in the anatomically-defined atlas. Identical minimization and prediction methods were run for each test dataset as well, but these results were not used to measure the accuracy or effectiveness of the prediction methods.

## Cortical magnification

Cortical magnification was calculated using both the observed retinotopic maps and the inferred maps that were produced by combining each subject's full retinotopy dataset with the retinotopic prior. This combination of data should, in theory, produce the highest-quality retinotopic map predictions of which we are capable (see Results and Discussion). Cortical magnification was calculated

by first projecting all vertices in a single visual area (such as V1) into the visual field based on their pRF centers. The cortical magnification of a particular polar angle and eccentricity is then the total white vertex surface-area (as calculated by FreeSurfer) of all pRF centers within a disk of some radius $\alpha$, divided by the area of the disk ($\pi\alpha^2$). For an eccentricity $\rho$, we used a radius $\alpha = \rho/3$.

## Acknowledgements

The research was supported by NIH grants R00-EY022116, R01-MH111417, R01- EY027964, and R01-EY027401. The authors declare no competing financial interests.

All figures and plots were produced using the Neurotica library (https://github.com/noahbenson/neurotica) for Mathematica 11 (*Wolfram Research, 2018*). Registrations and interpolations were performed with the assistance of the nibabel (DOI: 10.5281/zenodo.1287921), numpy (*Oliphant, 2006*), and scipy (*Jones, 2001*) libraries for Python.

## Additional information

### Funding

| Funder | Grant reference number | Author |
|---|---|---|
| National Eye Institute | R01 EY027964 | Noah C Benson<br>Jonathan Winawer |
| National Eye Institute | R01 EY027401 | Noah C Benson<br>Jonathan Winawer |
| National Institute of Mental Health | R01 MH111417 | Noah C Benson |
| National Eye Institute | R00-EY022116 | Jonathan Winawer |

The funders had no role in study design, data collection and interpretation, or the decision to submit the work for publication.

### Author contributions

Noah C Benson, Conceptualization, Resources, Data curation, Software, Formal analysis, Validation, Investigation, Visualization, Methodology, Writing—original draft, Writing—review and editing; Jonathan Winawer, Conceptualization, Resources, Data curation, Supervision, Funding acquisition, Investigation, Project administration, Writing—review and editing

### Author ORCIDs

Noah C Benson http://orcid.org/0000-0002-2365-8265
Jonathan Winawer http://orcid.org/0000-0001-7475-5586

### Ethics

Human subjects: This study was conducted with the approval of the New York University Institutional Review Board (IRB-FY2016-363) and in accordance with the Declaration of Helsinki. Informed consent was obtained for all subjects.

### Decision letter and Author response

Decision letter https://doi.org/10.7554/eLife.40224.023
Author response https://doi.org/10.7554/eLife.40224.024

## Additional files

### Supplementary files

• Supplementary file 1. Cross-validation schema. To evaluate the accuracy of the predictions of retinotopic maps, we employ a cross-validation schema. Each subject's 12 retinotopic mapping scans were divided into one large set of validation data as well as 21 smaller sets of training data. An

additional dataset of all 12 scans was used for analysis of retinotopic properties not linked to evaluation of the quality of the predicted maps.

DOI: https://doi.org/10.7554/eLife.40224.014

• Supplementary file 2. Deriving the anatomically-defined atlas of retinotopy (the prior). (A) The group-average polar angle (top) and eccentricity (bottom) maps. The cortical surface is inflated to a sphere then flattened to a map. (B) The model of retinotopy shown with polar angle plotted on the left and eccentricity plotted on the right hemispheres. (C) The retinotopic prior is constructed from the group-average data using an updated version of the method described by *Benson et al. (2014)*. Note that while only eccentricity is shown, polar angle and eccentricity are registered simultaneously. The checkerboard underlay illustrates the anatomical warping. (D) There is approximate agreement between the boundaries of visual areas V1, V2, and V3 as defined by two atlases. The Wang et al. maximum probability atlas (2015) and the retinotopic template defined here have similar boundaries. The template extends from 0° to 90° eccentricity, whereas the Wang et al atlas is limited to the field of view of their experiments (14°), hence the template maps are larger. (E) Because there is a topological isomorphism between the cortical surface, the left and right hemifields, and the model of retinotopy, the three representations have exact one-to-one correspondences.

DOI: https://doi.org/10.7554/eLife.40224.015

• Supplementary file 3. The retinotopic prior. (A) The 181-subject group-average retinotopic maps from the Human Connectome Project 7T Retinotopy Dataset are shown. These maps were used to construct the prior. Black arrows in the left-most plots indicate 'notches' of the V3 representation of the upper and lower vertical meridians that are absent in the group-average data. (B) The retinotopic prior is shown from 0 to 12° of eccentricity with boundary lines between areas. All 12 retinotopic areas included in the prior are shown.

DOI: https://doi.org/10.7554/eLife.40224.016

• Supplementary file 4. Warp fields in the V1-V3 region across all subjects. The warp fields are calculated using the individual vertex deviations during the registration process (*Figure 3C*). The top row shows the mean vertex deformation across all subjects while the bottom three rows show the first three principal components of the deviations. The brightness of the arrows is based on their relative lengths. Note that because the top row shows the mean warp-field across subjects, the exact direction of the arrows is significant; however, in the bottom three rows, the principal component axes are shown, so the inversion of the arrows is equivalent to the plotted arrows.

DOI: https://doi.org/10.7554/eLife.40224.017

• Supplementary file 5. Retinotopic maps for subjects (A) 198653 and (B) 644246 from the Human Connectome Project. These two subjects have unusual retinotopic organization in the polar angle maps of their left hemispheres (A) and on both hemispheres (B); this organization is not accounted for my our model of retinotopy and thus provides an example of how our Bayesian inference method performs when provided with atypical retinotopic maps. In the polar angle maps (top), black lines indicate V1/V2/V3 boundaries. In the eccentricity maps (bottom), black lines show the outer V3 boundaries and the 0.5°, 1°, 2°, 4° and 8° iso-eccentricity curves. Black arrows indicate the sites of atypical retinotopic organization.

DOI: https://doi.org/10.7554/eLife.40224.018

• Transparent reporting form

DOI: https://doi.org/10.7554/eLife.40224.019

### Data availability

All data generated or analyzed in this study have been made public on an Open Science Foundation website: https://osf.io/knb5g/. Preprocessed MRI data as well as analyses and source code for reproducing figures and performing additional analyses can be found on the Open Science Foundation website https://osf.io/knb5g/. Performing Bayesian inference using your own retinotopic maps. To perform Bayesian inference on a FreeSurfer subject, one can use the neuropythy Python library (https://github.com/noahbenson/neuropythy). For convenience, this library has also been packaged into a Docker container that is freely available on Docker Hub (https://hub.docker.com/r/nben/neuropythy). The following command will provide an explanation of how to use the Docker: '> docker run -it –rm nben/neuropythy:v0.5.0 register_retinotopy –help'. Detailed instructions on how to use

the tools documented in this paper are included in the Open Science Foundation website mentioned above.

The following dataset was generated:

| Author(s) | Year | Dataset title | Dataset URL | Database and Identifier |
|---|---|---|---|---|
| Benson NC, Winawer J | 2018 | Bayesian Models of Human Retinotopic Organization | https://osf.io/knb5g/ | Open Science Framework, osf.io/knb5g/ |

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
