## [Decision Letter]

Thank you for submitting your article "Bayesian Analysis of Retinotopic Maps" for consideration by *eLife*. Your article has been reviewed by three peer reviewers, including Mark Schira as the Reviewing Editor and Reviewer #3, and the evaluation has been overseen by Joshua Gold as the Senior Editor. The following individual involved in review of your submission has agreed to reveal their identity: D. Samuel Schwarzkopf (Reviewer #1).

The reviewers have discussed the reviews with one another and the Reviewing Editor has drafted this decision combining the comments from all three reviewers to help you prepare a revised submission.

Summary:

The manuscript by Benson and Winaver introduces a very sophisticated toolbox that combines a prior with fMRI scanning to estimate the retinotopic layout of a large set of visual areas for an individual subject. This work is an extension of previous work from the first author. The authors previous work has proven very successful and useful, and there is little doubt that the current work will be very useful, too. These techniques could theoretically be very beneficial in situations where only minimal data can be collected (elderly, children, patients). Since these Bayesian maps also contain automatic delineation, they could be very useful for reducing experimenter biases in manual delineation, which is still the standard to the field. Of course, this comes with the caveat of reduced flexibility, a potential systematic bias and that the delineation of many areas beyond V3 remains controversial. The auto-delineation is only as good as the atlas used as prior – but even then, the atlas procedure would enhance the reproducibility of findings. In general, this method is solid, cutting edge, and provides a significant improvement over previous procedures.

Essential revisions:

The overall assessment of the manuscript from all three reviewers is quite positive, but they raise a set of concerns, as follows.

1) There were questions about how well the methods can deal with extreme individual variability. As it stands, the manuscript presents a detailed and informative analysis of variability in the anatomical and functional features of visual field maps. The ability to characterize the extent to which variations in retinotopic organization are due to anatomical differences versus differences in the structure-function relationship is remarkable. However, the discussion on how this method would allow the quantification of more extreme and unexpected departures from expected topographical organization should be expanded. There exist several reports showing how clinical conditions (for example: achiasma, hemi-hydranencephaly and albinism) can affect the topographic organization of areas in early visual cortex. It should be discussed what results the method would provide if presented with such anomalies, expected or unexpected. At the present time there is a brief suggestion in the introduction about clinical applications, but we believe it should be expanded in the Discussion.

2) On a related note, the Bayesian inference maps do a good job at revealing individual differences in the functional architecture of visual cortex. However, while the deviations from a validation data set are small for these Bayesian maps, it would be informative to understand under what situations this technique *fails*. It is unsurprising that the noisy “Data alone” or the “Anatomy alone” maps contain errors – but could the authors elaborate on any systematic errors the Bayesian method has? Do most errors originate in the prior or the empirical retinotopic maps?

3) In a proper Bayesian formulation, the effect of the evidence on the posterior should depend on both the strength of the evidence but also on the certainty of the prior, which would presumably vary with the visual area. Our understanding of the methods was that the prior was essentially weighted equally at all locations. However, the certainty of the prior maps is arguably better for V1 than the higher areas (this is more of a problem for the higher regions beyond V3 and the authors already acknowledge that this aspect is still experimental).

Moreover, there was some concern that, as more data are introduced the estimated maps do not seem to improve much further. We understand and appreciate the comments from the authors that they believe this as a strength rather than a bug, because even excessive amount of data might not hold the ground truth. However, it still is a concern: under Bayesian optimization, the uncertainty of the data estimate should continue to get smaller with increasing repetition. This does not seem to be the case, instead after a small amount of data the model 'refuses' to make any further compromises. Further discussion of this point might be useful.

4) The authors should be much clearer and more upfront what the novel contribution of this work is, as opposed to previous work. From our judgment the improvements are substantial, but the reader has to piece together what is based on previous work and what novel, hence a succinct rapport early on would be much appreciated. Along a similar line, we believe the manuscript would benefit from being made more concise.

5) The authors already discuss this to some degree, but of course there is no proper ground truth in these analyses. They used a large retinotopic data set as validation data, which seems entirely reasonable. But this has some implications for the individual differences in functional architecture that this analysis reveals: offsets between cortical folding structure and retinotopic organisation could be because of errors in alignment between the functional scans and the cortical sampling. Even a small offset could masquerade as a shift in visual field borders. Thus, some differences may reveal not individual differences in maps but in the idiosyncrasies in image alignment.

6) We were a bit confused about the pRF size data. Both the pattern and magnitude of results in the Bayesian inference map don't match the empirical pRF data very well. pRF sizes in the actual data are much greater and differ systematically between V1, V2, and V3. The Bayesian data have smaller pRFs (intercept at 0) and V1 and V2 are essentially indistinguishable. The authors should discuss those discrepancies and other issues that might affect pRF size, such as fixation stability and the size of the centre-surround configuration.

Moreover, the model predicts visual field positions not included in the training data and returns an estimate of pRF size. While the fit between the inferred and the validated eccentricity is remarkable, the plot on the wide-field data is based on a single participant. This limitation should be stressed in the text as a cautionary note. Extrapolation is a notoriously hard problem. The quality of the data plays a major role. A naïve participant, with not-so-stable fixation, and maybe moving in the scanner could result in a different extrapolation trend.

7) Inter-individual differences that remain after the first anatomical alignment step are characterised as variability in the structure vs. function relation, and they may or may not largely be that. However, no alignment process is optimal and some of the remaining variance is undoubtedly due to insufficiencies of the anatomical alignment process. The alignment process used is based on the nice work from Hinds et al., but one does wonder if the large dataset used may not allow improvements to this process – and if only through fine tuning of some parameters.

8) Figure 9 is very hard to decipher, it needs more space and thinner lines and different rendering of the error range. It also seems to suggest that the magnification curves of V2 and V3 cross that of V1 at around 1.5-3 degrees. Crossing of magnification curved was first shown by Schira et al. (2009) and explained by Schira et al. (2010). Your data seems to suggest the crossing is at higher eccentricities than the 0.7-1 degree reported by Schira et al., which is possible but deserves some discussion. Specifically, since measured and inferred maps disagree on this value.

---

## [Author Response]

Essential revisions:The overall assessment of the manuscript from all three reviewers is quite positive, but they raise a set of concerns, as follows.1) There were questions about how well the methods can deal with extreme individual variability. As it stands, the manuscript presents a detailed and informative analysis of variability in the anatomical and functional features of visual field maps. The ability to characterize the extent to which variations in retinotopic organization are due to anatomical differences versus differences in the structure-function relationship is remarkable. However, the discussion on how this method would allow the quantification of more extreme and unexpected departures from expected topographical organization should be expanded. There exist several reports showing how clinical conditions (for example: achiasma, hemi-hydranencephaly and albinism) can affect the topographic organization of areas in early visual cortex. It should be discussed what results the method would provide if presented with such anomalies, expected or unexpected. At the present time there is a brief suggestion in the introduction about clinical applications, but we believe it should be expanded in the Discussion.

The reviewers are correct that the manuscript emphasized retinotopy measures and models for typical subjects, and did not address how the method would work for unusual or extreme cases. This is an excellent suggestion, and we have now added new data, analyses, and discussion on this issue. Specifically, there is a new section of the manuscript explicitly discussing unusual retinotopic maps and the Bayesian-inferred maps that our method finds when examining them. The new section is called, ‘Limitations and biases of the inferred maps.’ In brief, we used our method to infer the retinotopic maps of two subjects from the Human Connectome Project with unusual map organization. These unusual cases were noted and discussed by Van Essen and Glasser (2018) in a recent review, based on data from a manuscript we collaborated on with them, currently in press (Benson et al., 2018). Both of these subjects have unusual topologies in the functional organization of their left hemisphere dorsal V3 maps that are inconsistent with the retinotopic model we use and the rest of the field generally assumes. Although we find that the inferred maps for these subjects have a different topology than the subjects’ observed maps, the inferred maps appear to be reasonable attempts to explain the observed retinotopic organization with a model that does not topologically match the data.

These maps and model fits are shown in the new Supplementary file 5, and the expanded discussion of the limits and potential applications to clinical populations can be found in the main text (‘Limitations and biases of the inferred maps.’).

*2) On a related note, the Bayesian inference maps do a good job at revealing individual differences in the functional architecture of visual cortex. However, while the deviations from a validation data set are small for these Bayesian maps, it would be informative to understand under what situations this technique* fails*. It is unsurprising that the noisy “Data alone” or the “Anatomy alone” maps contain errors – but could the authors elaborate on any systematic errors the Bayesian method has? Do most errors originate in the prior or the empirical retinotopic maps?*

How and why the Bayesian maps produce systematic errors is an important question, but it is difficult if not impossible to answer definitively due to a lack of ground truth maps. Nonetheless, the reviewers are correct that there are some systematic deviations between the maps predicted from our method and our best measurements (validation data) that likely reflect more than measurement noise. For example, our template assumes that the polar angle map at the boundary between ventral V3v and hV4 reverses at precisely the lower vertical meridian. However, in many subjects’ retinotopic maps, this reversal is several degrees away from the meridian, such that there are few to no pRF centers within a few degrees of vertical. Because our Bayesian maps warp the positions of the vertices but not the retinotopic values of the anatomically-defined template, and because this template contains a complete representation of the visual field (all the way to the vertical meridian in each map), the vertical meridian always remains in the inferred maps. This differs from the data, where the vertical meridian is often missing. Whether this is a bug or a feature depends on properties of ground truth maps, which are currently not perfectly known.

This example is discussed in the third paragraph of the new section of the paper on the limitations and biases of the method. See also the related response to point 1 above.

3) In a proper Bayesian formulation, the effect of the evidence on the posterior should depend on both the strength of the evidence but also on the certainty of the prior, which would presumably vary with the visual area. Our understanding of the methods was that the prior was essentially weighted equally at all locations. However, the certainty of the prior maps is arguably better for V1 than the higher areas (this is more of a problem for the higher regions beyond V3 and the authors already acknowledge that this aspect is still experimental).

We agree that the certainty of the prior maps could (and surely does) differ across cortical locations. We did not incorporate a variable level of certainty into our model. A more complete model would do so. This omission does not, however, indicate that our model is not a proper Bayesian formulation; rather it means that the prior probability distributions we assumed differ from the actual distributions. Consider the distance penalty. Our registration algorithm penalizes model solutions as the distance between neighboring vertices increases. The size of the penalty depends on the distance but not on *which* vertices are moved in the registration. This is akin to a prior probability distribution of edge lengths that is the same for all edges. This assumption is simple to implement and describe, though surely incorrect in detail. For example, some edges are unlikely to change much during registrations, and others are likely to change a lot. Relaxing this assumption would not require a conceptual change to the model, but would require an implementation change: an edge-specific cost function rather than the same cost function for all edges. Deriving independent probability distributions (and hence corresponding cost functions) for each edge would likely improve the model, but is currently beyond the scope of this paper.

We believe there is a value in simplified assumptions (such as identical probability distributions for all edges) even at the cost of accuracy. That said, in ongoing work, we are in fact trying to derive edge-specific distributions from a large retinotopy dataset (Benson et al., 2018).

We now discuss this issue in the three paragraphs beginning after Equation 1 in the section, ‘Making the Bayesian inference explicit.’ Moreover, we plot the mean warp field, and the first several principal components of the warp fields. These warp fields show how much (and in what direction) the vertices are shifted from the prior by the Bayesian method (Supplementary file 4). The results confirm the reviewers’ assumption: some vertices have high loadings on the PCs, and others do not, indicating that the Bayesian fits result in some vertices being shifted much more than others.

Moreover, there was some concern that, as more data are introduced the estimated maps do not seem to improve much further. We understand and appreciate the comments from the authors that they believe this as a strength rather than a bug, because even excessive amount of data might not hold the ground truth. However, it still is a concern: under Bayesian optimization, the uncertainty of the data estimate should continue to get smaller with increasing repetition. This does not seem to be the case, instead after a small amount of data the model 'refuses' to make any further compromises. Further discussion of this point might be useful.

It is true that the template solutions are relatively stable with a small amount of data. One could view this as a success or a failure. There are two reasons for the stability. One has to do with the prior probability distributions and one has to do with the likelihood.

The prior: The topology of the template is preserved by a hard rule. This is now discussed explicitly in terms of prior probability solutions. If a particular map configuration has a prior probability of 0, then this configuration will never be the Bayesian solution, no matter how precise the measurement. This puts an upper bound on how accurate the Bayesian models can be.

The likelihood: The likelihood in the Bayesian formulation does in fact depend on measurement precision. Specifically, the cost function grows with distance between a vertex and its anchor point, and with the variance explained by the pRF model (Equation 2, row 4, *w* term). As a result, vertices with higher quality data contribute more to the Bayesian fit. However, this growth is linear with respect to the coherence of the pRF model. Because we do not know the mapping between model coherence and the true likelihood, our weights are effectively chosen by heuristic.

We now make these points explicit in the text. See the next-to-last paragraph of the section, ‘The inferred maps make highly accurate predictions using very little data.’ as well as the section, ‘‘Making the Bayesian inference explicit.’ See also our response to query points 2 and 3.

4) The authors should be much clearer and more upfront what the novel contribution of this work is, as opposed to previous work. From our judgment the improvements are substantial, but the reader has to piece together what is based on previous work and what novel, hence a succinct rapport early on would be much appreciated. Along a similar line we believe the manuscript would benefit from being made more concise.

Agreed. We have reworked sections of the Introduction to make the novel contributions much clearer up-front and have cleaned up our writing throughout.

5) The authors already discuss this to some degree, but of course there is no proper ground truth in these analyses. They used a large retinotopic data set as validation data, which seems entirely reasonable. But this has some implications for the individual differences in functional architecture that this analysis reveals: offsets between cortical folding structure and retinotopic organisation could be because of errors in alignment between the functional scans and the cortical sampling. Even a small offset could masquerade as a shift in visual field borders. Thus, some differences may reveal not individual differences in maps but in the idiosyncrasies in image alignment.

We agree entirely that the image-alignment errors are a potential source of noise and should therefore temper our conclusions. We have added additional text discussing this issue. See updated text in the second-to-last paragraph of the section, ‘Individual differences in the V1-V3 structure-function relationship across subjects are substantial.’

6) We were a bit confused about the pRF size data. Both the pattern and magnitude of results in the Bayesian inference map don't match the empirical pRF data very well. pRF sizes in the actual data are much greater and differ systematically between V1, V2, and V3. The Bayesian data have smaller pRFs (intercept at 0) and V1 and V2 are essentially indistinguishable. The authors should discuss those discrepancies and other issues that might affect pRF size, such as fixation stability and the size of the centre-surround configuration.

In the previous version, we compared our pRF sizes to those from a prior paper (Kay et al., 2013), in which the V1 and V2 pRF sizes did not differ substantially.

We removed the comparison to the Kay et al., 2013 paper (Figure 10) and clarified the description of pRF size (see the section “The Bayesian inferred maps accurately reproduce systematic properties of the visual field maps”).

Moreover, the model predicts visual field positions not included in the training data and returns an estimate of pRF size. While the fit between the inferred and the validated eccentricity is remarkable, the plot on the wide-field data is based on a single participant. This limitation should be stressed in the text as a cautionary note. Extrapolation is a notoriously hard problem. The quality of the data plays a major role. A naïve participant, with not-so-stable fixation, and maybe moving in the scanner could result in a different extrapolation trend.

Agreed. We now note that one of the two panels on extrapolation of eccentricity comes from only one subject, and that the accuracy of extrapolation to untested eccentricities likely depends on the quality of the measured data and subject fixation (see the section “The Bayesian model accurately predicts visual field positions not included in the training data”.

7) Inter-individual differences that remain after the first anatomical alignment step are characterised as variability in the structure vs. function relation, and they may or may not largely be that. However, no alignment process is optimal and some of the remaining variance is undoubtedly due to insufficiencies of the anatomical alignment process. The alignment process used is based on the nice work from Hinds et al., but one does wonder if the large dataset used may not allow improvements to this process – and if only through fine tuning of some parameters.

Inter-subject cortical surface alignment remains a challenging problem and a source of uncertainty in our data. As we mention in the paper, it is conceivable that our method does little more than correct for insufficient cortical surface alignment; though we believe this not to be the complete story. In this paper we have chosen to employ FreeSurfer’s alignment tools due in part to their popularity in the community. This effectively makes it easier for our tools to be used to analyze existing datasets. We agree that large neuroimaging databases (e.g., the Human Connectome Project) engender a substantial opportunity to improve cortical alignment algorithms.

We have expanded our discussion of these ideas to the text (last two paragraphs of ‘Individual differences in the V1-V3 structure-function relationship across subjects are substantial.’).

8) Figure 9 is very hard to decipher, it needs more space and thinner lines and different rendering of the error range. It also seems to suggest that the magnification curves of V2 and V3 cross that of V1 at around 1.5-3 degrees. Crossing of magnification curved was first shown by Schira et al. (2009) and explained by Schira et al. (2010). Your data seems to suggest the crossing is at higher eccentricities than the 0.7-1 degree reported by Schira et al., which is possible but deserves some discussion. Specifically, since measured and inferred maps disagree on this value.

Agreed. Figure 9 (now Figure 10) has been cleaned up and replotted. The previous version of the figure used the standard error of the parameters rather than the standard error of the model fits themselves, and correcting this has made the plots more legible. We have additionally expanded our discussion of the cortical magnification regarding the crossover of cortical magnification of V1-V3.